# A mathematical theory of cooperative communication

**Pei Wang**
Rutgers University–Newark
peiwang@rutgers.edu

**Junqi Wang**
Rutgers University–Newark
junqi.wang@rutgers.edu

**Pushpi Paranamana**
Rutgers University–Newark
pushpi.paranamana@nd.edu

**Patrick Shafto**
Rutgers University–Newark
patrick.shafto@gmail.com

## Abstract

Cooperative communication plays a central role in theories of human cognition, language, development, culture, and human-robot interaction. Prior models of cooperative communication are algorithmic in nature and do not shed light on why cooperation may yield effective belief transmission and what limitations may arise due to differences between beliefs of agents. Through a connection to the theory of optimal transport, we establishing a mathematical framework for cooperative communication. We derive prior models as special cases, statistical interpretations of belief transfer plans, and proofs of robustness and instability. Computational simulations support and elaborate our theoretical results, and demonstrate fit to human behavior. The results show that cooperative communication provably enables effective, robust belief transmission which is required to explain feats of human learning and improve human-machine interaction.

## 1 Introduction

Cooperative communication is invoked across language, cognitive development, cultural anthropology, and robotics to explain people's ability to effectively transmit information and accumulate knowledge. Theories claim that people have evolved a specialized ecological niche [Tomasello, 1999, Boyd et al., 2011] and learning mechanisms [Csibra and Gergely, 2009, Grice, 1975, Sperber and Wilson, 1986], which explain our abilities to learn and accumulate knowledge; however, we lack mathematical theories that would allow us analyze basic properties of cooperative communication between agents.

Models of belief updating [Chater et al., 2008, Tenenbaum et al., 2011, Ghahramani, 2015] and action selection [Luce, 2012, Sutton et al., 1998] have recently been combined into models of cooperative communication in cognitive science [Shafto and Goodman, 2008a, Shafto et al., 2014], cognitive development [Eaves Jr et al., 2016, Bonawitz et al., 2011, Bridgers et al., 2019], linguistic pragmatics [Goodman and Stuhlmüller, 2013], and robotics [Ho et al., 2016, Hadfield-Menell et al., 2016, Fisac et al., 2017, Milli and Dragan, 2019]. These models are algorithms for computing cooperative communication plans using Theory of Mind reasoning. However, these models do not formalize the problem mathematically and therefore do not support general conclusions about the nature or limitations of cooperative communication.

Build upon mathematical and computational analysis, we provide answers to fundamental questions of cooperative communication. Our contributions are as follows. In Section 2, we interpret cooperative communication as a problem of optimal transport [Monge, 1781, Villani, 2008, Peyré and Cuturi, 2019], derive prior models of cooperative communication as special cases, and derive relationships to rate distortion theory. In particular, we theoretically guarantee the existence of optimal communication plan and algorithmically ensure the achievablility of such plans. In Section 3, we mathematically

analyze properties of cooperative communication including statistical interpretations, robustness to violations of common ground, and instability under greedy data selection. In Section 4, we computationally analyze robustness to common ground violations, sensitivity to greedy selection of data, approximate methods of correcting common ground, and demonstrate fit to human data.

## 2 Cooperative communication as a problem of optimal transport

Communication is a pair of processes considered between two agents, that we will refer to as a *teacher* and a *learner*, wherein the teacher selects data and the learner draws inferences based on those data. Optimal transport provides a mathematical framework for formalizing movement of one distribution to another, and therefore a framework for modeling communication. By recasting communication as belief transport we will gain access to mathematical and computational techniques for understanding and analyzing the problem of cooperative communication.

### 2.1 Background on Optimal Transport

Optimal Transport has been discovered in many settings and fields [Villani, 2008, Kantorovich, 2006, Koopmans, 1949, Dantzig, 1949, Brenier, 1991]. The general usefulness of optimal transport can be credited to the simplicity of the problem it solves. The original formulation, attributable to Monge [1781], involves minimizing the effort required to move a pile of dirt from one shape to another. Where Monge saw dirt, we may see any probability distribution.

**Entropy regularized Optimal Transport.** Formally, let $\mathbf{r} = (r_1, \ldots, r_n)$ and $\mathbf{c} = (c_1, \ldots, c_m)$ be probability vectors of length $n$ and $m$ respectively. A joint distribution matrix $P = (P_{ij})$ of dimension $n \times m$ is called a **transport plan**[1] between $\mathbf{r}$ and $\mathbf{c}$ if $P$ has $\mathbf{r}$ and $\mathbf{c}$ as its marginals. Denote the set of all *transport plans* between $\mathbf{r}$ and $\mathbf{c}$ by $U(\mathbf{r}, \mathbf{c})$. Further, let a non-negative $C = (C_{ij})_{n \times m}$ be the cost matrix, where $C_{ij}$ measures the cost of transportation between $r_i$ and $c_j$.

Cuturi [2013] proposed *Entropy regularized Optimal Transport (EOT)*. EOT seeks an optimal transport plan $P^{(\lambda)}$ that minimizes the entropy regularized cost of transporting $\mathbf{r}$ into $\mathbf{c}$. For a parameter $\lambda > 0$,

$$P^{(\lambda)} = \underset{P \in U(\mathbf{r},\mathbf{c})}{\arg\min} \ \{\langle C, P \rangle - \frac{1}{\lambda} H(P)\}, \tag{1}$$

where $\langle C, P \rangle = \sum_{i \in \mathcal{D}, j \in \mathcal{H}} C_{ij} P_{ij}$ is the Frobenius inner product between $C$ and $P$, and $H(P) := \sum_i^n \sum_j^m P_{ij} \log P_{ij}$ is the *entropy* of $P$. $P^{(\lambda)}$ is called a **Sinkhorn plan** with parameter $\lambda$.

**Sinkhorn scaling.** Sinkhorn plans can be computed efficiently via Sinkhorn scaling with linear convergence [Knight, 2008]. $(\mathbf{r}, \mathbf{c})$-*Sinkhorn scaling (SK)* [Sinkhorn and Knopp, 1967] of a matrix $M$ is simply the iterated alternation of row normalization of $M$ with respect to $\mathbf{r}$ and column normalization of $M$ with respect to $\mathbf{c}$ (See Example A.1 in Supplementary Text). When marginal distributions are uniform, we sometimes call it *Sinkhorn iteration*. It is shown in Cuturi [2013] that,

**Proposition 1.** *Given a cost matrix $C$, a Sinkhorn plan $P^{(\lambda)}$ of transporting $\mathbf{r}$ into $\mathbf{c}$ can be obtained by applying $(\mathbf{r}, \mathbf{c})$-Sinkhorn scaling on $P^{[\lambda]}$, where matrix $P^{[\lambda]}$ is defined by $P_{ij}^{[\lambda]} = e^{-\lambda \cdot C_{ij}}$, thus:*

$$P^{(\lambda)} = SK(P^{[\lambda]}) \quad and \quad P^{[\lambda]} := e^{-\lambda \cdot C} = (e^{-\lambda \cdot C_{ij}})_{n \times m}. \tag{2}$$

Much more is known about EOT and SK (see [Idel, 2016] and our Supplemental Text Section A).

### 2.2 Cooperative communication as optimal transport

*Cooperative communication* formalizes a single problem comprised of interactions between two processes: action selection (teaching) and inference (learning) [Shafto et al., 2014, Jara-Ettinger et al., 2016, Goodman and Frank, 2016, Fisac et al., 2017]. The teacher and learner have beliefs about hypotheses, which are represented as probability distributions. The process of teaching is to select data that move the learner's beliefs from some initial state, to a final desired state. The process of learning is then, given the data selected by the teacher, infer the beliefs of the teacher. The teacher's selection and

learner's inference incur costs. The agents minimize the cost to achieve their goals. Communication is successful when the learner's belief, given the teacher's data, is moved to the target distribution. The connection between EOT and cooperative communication is established by modeling each process, teaching and learning, as a classical EOT problem.

**Framework.** Let $\mathcal{H}$ be a hypothesis space and $\mathcal{D}$ be a data space. Denote the common ground between agents: the shared priors on $\mathcal{H}$ and $\mathcal{D}$ by $P_0(\mathcal{H})$ and $P_0(\mathcal{D})$, the shared initial matrix over $\mathcal{D}$ and $\mathcal{H}$ by $M$ of size $|\mathcal{D}| \times |\mathcal{H}|$. In general, up to normalization, $M$ is simply a non-negative matrix which also specifies the consistency between data and hypotheses [2]

In cooperative communication, a teacher's goal is to minimize the cost of transforming the shared prior over hypotheses $P_0(\mathcal{H})$ into shared prior over data points $P_0(\mathcal{D})$. We define the teacher's cost matrix $C^T = (C_{ij}^T)_{|\mathcal{D}| \times |\mathcal{H}|}$ as:

$$C_{ij}^T = -\log P_L(h_j|d_i) + S_T(d_i), \tag{3}$$

where $P_L(h_j|d_i)$ is the learner's likelihood of inferring hypothesis $h_j$ given data $d_i$, and $S_T(d_i)$ is determined by the teacher's prior on the data $d_i$ which can be interpreted as teacher's expense of selecting data $d_i$. Thus, taking cooperation into consideration, data $d$ is good for a teacher who wishes to communicate $h$ if $d$ has a low selecting expense and the learner assigns a high probability to $h$ after updating with $d$. Symmetrically, a learner's cost matrix $C^L = (C_{ij}^L)_{|\mathcal{D}| \times |\mathcal{H}|}$ is defined as $C_{ij}^L = -\log P_T(d_i|h_j) + S_L(h_j)$, where $P_T(d_i|h_j)$ is the teachers's likelihood of choosing data $d_i$ given hypothesis $h_j$ and $S_L(h_j)$ is determined by the learner's prior on the hypothesis $h_j$.

**Optimal Planning.** A *teaching plan* is a joint distribution $T = (T_{ij})$ over $\mathcal{D}$ and $\mathcal{H}$, where each element $T_{ij} = P_T(d_i, h_j)$ represents the probability of the teacher selecting $d_i$ to convey $h_j$. Similarly a *learning plan* is a joint distribution $L = (L_{ij})$, where $L_{ij} = P_L(d_i, h_j)$ represents the probability of the learner inferring $h_j$ given $d_i$. Column normalization of $T$ and row normalization of $L$ are called *conditional communication plans*.

Under our framework, the *optimal cooperative communication plans* that minimize agents' costs on transmitting between $\mathcal{H}$ and $\mathcal{D}$ are precisely the *Sinkhorn plans* as in Equation (1). Hence, as a direct application of Proposition 1, we have

**Proposition 2.** *Optimal cooperative communication plans, $T^{(\lambda)}$ and $L^{(\lambda)}$, that achieve Sinkhorn plans of EOT with given $\lambda$, can be obtained through Sinkhorn Scaling on matrices determined by the common ground between agents: priors $P_0(\mathcal{H})$, $P_0(\mathcal{D})$ and shared consistency matrix $M$.*

Construction of optimal plans $T^{(\lambda)}$ and $L^{(\lambda)}$ using Prop. 2 is illustrated as follows. Assume zero expense of data selection and uniform priors on both $\mathcal{D}$ and $\mathcal{H}$. A natural estimation of the learner is a naive learner whose learning plan is fully based on the shared $M$. In this case, the teacher may approximate the learner's likelihood matrix by $L_0$, the row normalization of $M$. Hence the teacher's cost matrix defined in Eq.(3) has the form $C^T = -\log L_0$. As in Eq.(2), the optimal teaching plan with regularizer $\lambda$, denoted by $T^{(\lambda)}$, can be obtained by applying Sinkhorn iterations on $T^{[\lambda]}$, i.e.

$$T^{(\lambda)} = SK(T^{[\lambda]}) = SK(e^{-\lambda \cdot C^T}) = SK(e^{\lambda \cdot \log L_0}) = SK(L_0^{[\lambda]}), \tag{4}$$

where $L_0^{[\lambda]}$ represents the matrix obtained from $L_0$ by raising each element to the power of $\lambda$. Symmetrically, the optimal learning plan with regularizer $\lambda$, denoted by $L^{(\lambda)}$, can be reached by Sinkhorn iteration on $L^{[\lambda]} = e^{-\lambda \cdot C^L} = T_0^{[\lambda]}$, where $T_0$ is the column normalization of $M$. Parameter $\lambda$ controls the agents' greediness towards deterministic plans, which is investigated in Section 3.3.

## 2.3 Unifying existing theories of cooperative communication

A wide range of existing cooperative models in pragmatic reasoning, social cognitive development and robotics can be unified as approximate inference for EOT. The major variations among these models are: depth of Sinkhorn scaling and choice of parameter $\lambda$. See a brief summary in Table 1

**Fully recursive Bayesian reasoning.** The first class is based on the classic Theory of Mind recursion, including *pedagogical reasoning* [Shafto and Goodman, 2008b, Shafto et al., 2012, 2014] and

Table 1: Unifying existing cooperative models by EOT framework

| Example of Existing Models | Depth of SK | choice of $\lambda$ | Stochasticity |
|---|---|---|---|
| Pedagogical Reasoning [Shafto et al., 2014] | until converge | fit per data | probabilistic |
| Cooperative Inference [Yang et al., 2018] | until converge | 1 | probabilistic |
| Bayesian Teaching [Eaves Jr et al., 2016] | 1 step | 1 | probabilistic |
| Machine Teaching [Zhu, 2013] | 1 step | N.A. (argmax) | deterministic |
| Naive Utility Calculus [Jara-Ettinger et al., 2016] | 1 step | 1 | probabilistic |
| RSA [Goodman and Frank, 2016] | 1-2 steps | fit per data | probabilistic |
| Value Alignment [Fisac et al., 2017] | 1 step | fit per data | deterministic |

*cooperative inference* [Yang et al., 2018, Wang et al., 2019]. These models use fully Bayesian inference to compute the exact Sinkhorn plans (i.e. Sinkhorn scaling until convergence) for the case of $\lambda = 1$. In more detail, these models emphasize that agents' optimal conditional communication plans, $T^\star = P_T(\mathcal{D}|\mathcal{H})$ and $L^\star = P_L(\mathcal{H}|\mathcal{D})$ should satisfy the following system of interrelated equations, each of which is in form of the Bayes's rule:

$$P_L(h|d) = \frac{P_T(d|h)\,P_{L_0}(h)}{P_L(d)} \qquad P_T(d|h) = \frac{P_L(h|d)\,P_{T_0}(d)}{P_T(h)} \qquad (5)$$

where $P_L(d)$ and $P_T(h)$ are the normalizing constants. The main theorem in Yang et al. [2018] shows that assuming $P_{L_0}(h)$ and $P_{T_0}(d)$ are uniform priors over $\mathcal{H}$ and $\mathcal{D}$, Eq.(5) can be solved using SK iteration on the shared matrix $M$. Hence coincide with Sinkhorn plans of EOT. Moreover, benefiting directly from the EOT framework, Prop. 2 implies and extends this result to *arbitrary priors*:

**Proposition 3.** [3] *Optimal conditional communication plans, $T^\star$ and $L^\star$, of a cooperative inference problem with arbitrary priors, can be obtained through Sinkhorn scaling. In particular, as a direct consequence, cooperative inference is a special case of the unifying EOT framework with $\lambda = 1$.*

**One-step approximate inference.** The second class is based on human behaviors such as *Naive Utility Calculus* [Jara-Ettinger et al., 2016, Jern et al., 2017], *Rational Speech Act* (RSA) theory [Goodman and Frank, 2016, Franke and Jäger, 2016] and *Bayesian Teaching* [Eaves Jr and Shafto, 2016, Eaves Jr et al., 2016], and recent advances in robotics and machine learning, such as *machine teaching* [Zhu, 2013, 2015], *pedagogical interaction* [Ho et al., 2016, 2018] and *value alignment* [Hadfield-Menell et al., 2016, Fisac et al., 2017, Jara-Ettinger, 2019]. These models compute one or two steps of the Sinkhorn scaling, then approximate the Sinkhorn plans of EOT either with the resulting probability distribution or form a deterministic plan using argmax (See detailed demonstrations in Supplementary Text Sec. B). Greediness parameter $\lambda$ is fitted as hyperparameter for different applications. The EOT framework suggests in many cases, such approximations are far from optimal (illustrated in Fig. 1) and are much more sensitive to agents' estimation of the other agent (see Sec. 3.2).

## 2.4 Connections to Information theory

Cooperative communication, like standard information theory, involves communication over a channel. It is therefore interesting and important to ask whether there is a formal connection. The EOT formulation shows that the cooperative communication is closely related to lossy data compression in *rate-distortion theory* as follows.

Let $X = \{x_i\}_{i=1}^m$ be the source (input) space, $Y = \{y_j\}_{j=1}^n$ be the receiver (output) space, $P_0(X)$ be a fixed prior on $X$ and $Q = P(y_j|x_i)$ be a compression scheme. Denote the distortion between $x_i$ and $y_j$ by $d(x_i, y_j)$, which measures the cost of representing $x_i$ in terms of $y_j$. The **distortion** of a given compression scheme $Q$ is defined to be: $D_Q(X, Y) = \sum_{i,j} P_0(x_i) \cdot P(y_j|x_i) \cdot d(x_i, y_j) = \sum_{i,j} P(x_i, y_i) \cdot d(x_i, y_j)$. The amount of information (bits per symbol) communicated through scheme $Q$ is measured by the **mutual information**, $I(X, Y) = H(X) + H(Y) - H(X, Y)$, where

$H(X)$, $H(Y)$ and $H(X, Y)$ are entropy of $P_0(X)$, $P_0(Y)$ and $P(X, Y)$ respectively. The classical *Distortion-rate function*, formulates the problem of minimizing distortion while passing at most $R$-bit per input symbol of information, thus find:

$$Q^* = \arg\inf_Q D_Q(X, Y) \text{ subject to } I(X, Y) < R. \tag{6}$$

EOT minimizes the communication distortion by replacing the hard constraint on mutual information in Eq. (6) by a soft regularizer. Consider the case where $X = \mathcal{H}$, $Y = \mathcal{D}$, EOT is the problem that among all the compression scheme (communication plans) satisfying $P_0(\mathcal{H}) = \mathbf{c}$ and $P_0(\mathcal{D}) = \mathbf{r}$, find the optimal plan that minimizes the distortion subject to penalties on bits per symbol. The penalty level is controlled by $\lambda$. Thus, in the notation of rate-distortion theory, Eq. (1) of EOT is equivalent to: $P^{(\lambda)} = \arg\inf_{P \in U(\mathbf{r}, \mathbf{c})} D_P(\mathcal{H}, \mathcal{D}) + \frac{1}{\lambda} I(\mathcal{H}, \mathcal{D})$.

## 3 Analyzing models of cooperative communication

### 3.1 EOT is statistically and information theoretically optimal

Optimal cooperative plans of EOT solves *entropy minimization* with marginal constraints through Sinkhorn scaling. Let $M$ be a joint distribution matrix over $\mathcal{D}$ and $\mathcal{H}$. Denote the set of all possible joint distribution with marginals $\mathbf{r} = P_0(\mathcal{D})$ and $\mathbf{c} = P_0(\mathcal{H})$ by $U(\mathbf{r}, \mathbf{c})$. Consider the question of finding the approximation matrix $P^*$ of $M$ in $U(\mathbf{r}, \mathbf{c})$ that minimizes its relative entropy with $M$:

$$P^* = \arg\inf_{P \in U(\mathbf{r}, \mathbf{c})} D_{\text{KL}}(P||M), \text{where } D_{\text{KL}}(P||M) = \sum_{i,j} P_{ij} \ln \frac{P_{ij}}{M_{ij}}, \tag{7}$$

The $(\mathbf{r}, \mathbf{c})$-SK scaling of $M$ converges to $P^*$ if the limit exists [Csiszar, 1989, Franklin and Lorenz, 1989]. We therefore directly interpret cooperative communication under EOT as minimum discrimination information for *pairs* of interacting agents.

Sinkhorn scaling also arises naturally as a *maximum likelihood estimation*. Let $\widehat{P}$ be the empirical distribution of i.i.d. samples from a true underlying distribution, which belongs to a model family. Then the log likelihood of this sample set over a distribution $M$ in the model family is given by $n \cdot \sum_{ij} \widehat{P}_{ij} \log M_{ij}$, where $n$ is the sample size. Comparing with Eq. (7), it is clear that maximizing the log likelihood (so the likelihood) over a given family of $M$ is equivalent to minimizing $D_{\text{KL}}(\widehat{P}||M)$. When the model is in the exponential family, the maximum likelihood estimation of $M$ can be obtained through SK scaling with empirical marginals [Darroch and Ratcliff, 1972, Csiszar, 1989]. Therefore, EOT planning can also be viewed as the maximum likelihood belief transmission plan.

### 3.2 Robustness to violations of common ground

In EOT, for a fixed regularizer $\lambda$, optimal plans are obtained through SK scaling on a matrix determined by $M$ w.r.t. $\mathbf{r} = P_0(\mathcal{D})$ and $\mathbf{c} = P_0(\mathcal{H})$. This can be viewed as a map $\Phi$, from $(M, \mathbf{r}, \mathbf{c})$ to the SK limit, where the *Common ground* – priors $P_0(\mathcal{D})$ & $P_0(\mathcal{H})$, and mappings from beliefs to data, $M$ – represent the assumption that cooperating agents share knowledge of each others' beliefs. However, it is implausible (even impossible) for any two agents to have exactly the common ground. We now investigate differentiability of EOT. This ensures robustness of the inference where agents' beliefs and mappings from beliefs to data differ, which shows the viability of cooperative communication in practice.

Let $M^{\epsilon_1}$, $\mathbf{r}^{\epsilon_2}$ and $\mathbf{c}^{\epsilon_3}$ be vectors obtained by varying elements of $M$, $\mathbf{r}$ and $\mathbf{c}$ at most by $\epsilon_i$, where $\epsilon_i > 0$ quantifies the amount of perturbation. We show that:

**Proposition 4.** *For any non-negative shared $M$ and positive marginals $\mathbf{r}$ and $\mathbf{c}$, if $\Phi(M^{\epsilon_1}, \mathbf{r}^{\epsilon_2}, \mathbf{c}^{\epsilon_3})$ and $\Phi(M, \mathbf{r}, \mathbf{c})$ exist, then $\Phi(M^{\epsilon_1}, \mathbf{r}^{\epsilon_2}, \mathbf{c}^{\epsilon_3}) \to \Phi(M, \mathbf{r}, \mathbf{c})$ as $M^{\epsilon_1} \to M, \mathbf{r}^{\epsilon_2} \to \mathbf{r}, \mathbf{c}^{\epsilon_3} \to \mathbf{c}$.*

Continuity of $\Phi$ implies that small perturbations on $M, \mathbf{r}, \mathbf{c}$, yield close solutions for optional plans. Thus cooperative communicative plans are robust to deviations from common ground between agents (see demonstrations in Sec. 4.1). In particular, if agents empirically estimate relevant aspects of common ground, derived cooperative plans will stabilize as the sample size increases.

Moreover, deviations in common ground are repairable in EOT without recomputing communication plans. When restricted to positive distribution $M$, Luise et al. [2018] shows that $\Phi(M, \mathbf{r}, \mathbf{c})$ is in fact smooth on $\mathbf{r}$ and $\mathbf{c}$. We further prove that $\Phi$ is also smooth on $M$. Therefore, the following holds:

**Theorem 5.** [4] *Let $\mathcal{M}$ be the set of positive matrices of shape $|\mathcal{D}| \times |\mathcal{H}|$, representing all possible shared distributions, let $\Delta^+_{|\mathcal{D}|}$ and $\Delta^+_{|\mathcal{H}|}$ be the set of all positive prior distributions over $\mathcal{D}$ and $\mathcal{H}$, respectively. Then $\Phi : \mathcal{M} \times \Delta^+_{|\mathcal{D}|} \times \Delta^+_{|\mathcal{H}|} \to \mathcal{M}$ is $C^\infty$.*

Theorem 5 guarantees that the optimal plans obtained through SK scaling are infinitely differentiable. Gradient descent can be carried out via *Automatic Differentiation* as in Genevay et al. [2017]. We explicitly derive the gradient of $\Phi$ with respect to both marginals and $M$ analytically in Sec E.2 of Supp.Text. Based on the derived closed form, we demonstrate that EOT agents can reconstruct a better cooperative plan using linear approximation once they realized the deviation from the previously assumed common ground in Sec. 4.3. In human communication, common ground is often inferred as part of the communication process [Luise et al., 2018, Hawkins et al., 2018]. Thus, the differentiability and the gradient formula significantly increase the flexibility and practicality of the EOT framework.

### 3.3 Instability under greedy data selection

We now explore the effect of $\lambda$ on EOT plans. To simplify notation, we focus on square matrices, similar analysis applies for rectangular matrices using machinery developed in Wang et al. [2019].

**Definition 6.** Let $A = (A_{ij})$ be an $n \times n$ square matrix and $S_n$ be the set of all permutations of $\{1, 2, \ldots, n\}$. Given $\sigma \in S_n$, the set $D^A_\sigma$ of $n$-elements $\{A_{1,\sigma(1)}, \ldots, A_{n,\sigma(n)}\}$ is called a **diagonal** of $A$ determined by $\sigma$. If $A_{k\sigma(k)} > 0$ for all $k$, we say that $D^A_\sigma$ is **positive**. $D^A_\sigma$ is called a **leading diagonal** if the product $d^A_\sigma = \Pi^n_{i=1} A_{i,\sigma(i)}$, is the largest among all diagonals of $A$.

**Definition 7.** Let $A, B$ be two $n \times n$ square matrices and $D^A_\sigma$ and $D^A_{\sigma'}$ be two diagonals of $A$ determined by permutations $\sigma, \sigma'$. Denote the products of elements on $D^A_\sigma, D^A_{\sigma'}$ by $d^A_\sigma, d^A_{\sigma'}$. Then $\mathrm{CR}(D^A_\sigma, D^A_{\sigma'}) = d^A_\sigma / d^A_{\sigma'}$ is called the **cross-product ratio** between $D^A_\sigma$ and $D^A_{\sigma'}$. Further, let the diagonals in $B$ determined by the same $\sigma$ and $\sigma'$ be $D^B_\sigma$ and $D^B_{\sigma'}$. We say $A$ is **cross-ratio equivalent** to $B$, if $d^A_\sigma \neq 0 \iff d^B_\sigma \neq 0$ and $\mathrm{CR}(D^A_\sigma, D^A_{\sigma'}) = \mathrm{CR}(D^B_\sigma, D^B_{\sigma'})$ holds for any $\sigma, \sigma'$.

Given $M$, consider the EOT problem for the teacher (similarly, for the learner). Recall that, as in Eq. (4), the optimal teaching plan $T^{(\lambda)}$ is the limit of SK iteration of $L^{[\lambda]}_0$. Note that the limits of SK scaling on $L^{[\lambda]}_0$ and $M^{[\lambda]}$ (obtained from $L_0$ or $M$ by raising each element to power of $\lambda$) are the same as they are cross-ratio equivalent (shown in Wang et al. [2019]). Therefore to study the dynamics of $\lambda$ regularized EOT solutions, we may focus on $M^{[\lambda]}$ and its Sinkhorn limit $M^{(\lambda)}$.

One extreme is when $\lambda$ gets closer to zero. If $\lambda \to 0$, $M^{[\lambda]}_{ij} = (M_{ij})^\lambda \to 1$ for any nonzero element of $M$. Thus $M^{[\lambda]}$ converges to a matrix filled with ones on the nonzero entries of $M$, and $M^{(\lambda)}$ converges to matrix $r^T c$ if $M$ has no vanishing entries. Hence $M^{(\lambda)}$ reaches low communicative effectiveness as $\lambda$ goes to zero (demonstrated in Sec. 4.2 with Fig. 1(b-c)).

The other extreme is when $\lambda$ gets closer to infinity. In this case, assuming uniform priors, we show:

**Proposition 8.** *$M^{(\lambda)}$ concentrates around the leading diagonals of $M$ as $\lambda \to \infty$.*

As $\lambda \to \infty$, the number of non-zero elements in $M^{(\lambda)}$ decreases. In the case when $M$ has only one leading diagonal, as $\lambda \to \infty$, $M^{(\lambda)}$ converges to a diagonal matrix (up to permutation). Thus, it forms a bijection between $\mathcal{D}$ and $\mathcal{H}$, and achieves the highest effectiveness.

The value of $\lambda$ causes variations on cross-ratios of $M^{(\lambda)}$, which affects the model's sensitivity to violations of common ground. Since $M^{[\lambda]}$ and $M^{(\lambda)}$ are cross-ratio equivalent, $M^{(\lambda)}$ has the same cross-ratio as the shared $M$ only when $\lambda = 1$. $M^{(\lambda \neq 1)}$ either exaggerates or suppresses the cross-product ratios of $M$, depending on whether $\lambda$ is greater or less than 1. Hence, deviations on common ground are amplified by large $\lambda$, which reduces the communication effectiveness. Indeed, when deviation causes two agents have different leading diagonals in their estimations of $M$, their optimal plans will be completely mismatched as $\lambda \to \infty$ (See detail examples in Supp. Text Sec. C).

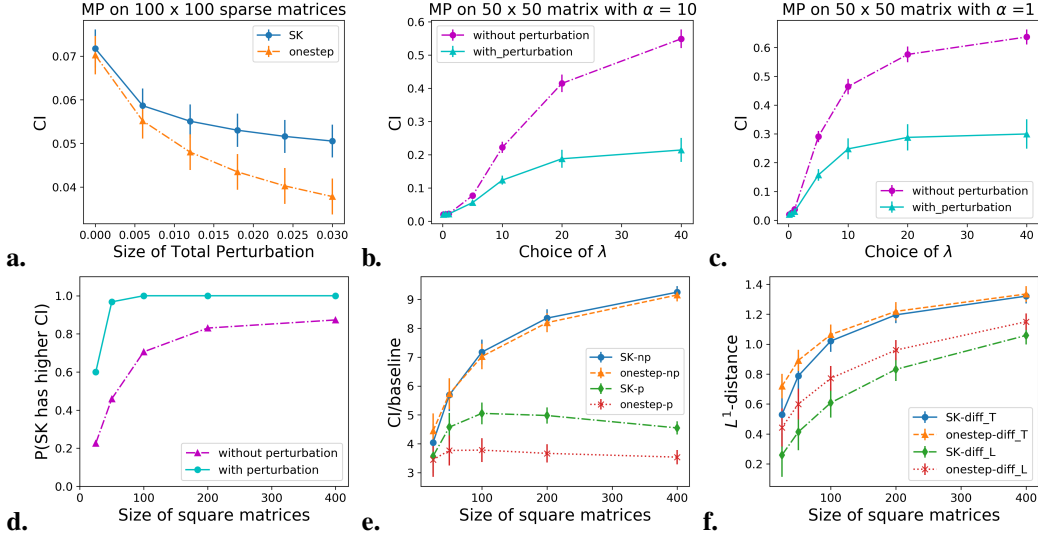

Figure 1: **a.** The Cooperative Index (CI) of Sinkhorn planning (SK) and its one step approximation (onestep) as total perturbation increases. **b-c.** The average CI of Sinkhorn planning for $50 \times 50$ matrices as $\lambda$ varies. **d-f.** $r = 0.03$, $\epsilon = 1$, dimension of $M$ varies as shown in $x$-axis. **d.** The probability that CI of SK planning is higher than its one step approximation. **e.** The average communication effectiveness for SK and onestep with and without perturbations denoted by SK-p, onestep-p, SK-np, onestep-np accordingly. **f.** The average difference of the teacher's (and learner's) SK plans (and one-step approx.) before and after perturbations, measured by the $L^1$-distance.

## 4 Experiments

We will now further illustrate properties of EOT through simulations. Effectiveness of communication will be measured via the **Cooperative Index (CI)** $\mathrm{CI}(T, L) := \frac{1}{|\mathcal{H}|} \sum_{ij} L_{ij} T_{ij}$ [Yang et al., 2018]. It ranges between $0$ and $1$ and measures the communication effectiveness of a pair of plans $T$ and $L$. Intuitively, $\mathrm{CI}(T, L)$ quantifies the effectiveness as the average probability that a hypothesis can be correctly inferred by a learner given the teacher's selection of data.

### 4.1 Perturbation on common ground

In this section, we stimulate perturbations by Monte Carlo method to compare the robustness of the Sinkhorn planning and its one-step approximation.

**Basic Set-Up.** Assume a uniform prior on $\mathcal{D}$ and $\lambda = 1$. Shared matrix $M$ and prior over $\mathcal{H}$ are sampled from symmetric Dirichlet distribution with hyperparameter $\alpha = 0.1$[5]. Sample size is $10^6$ per plotted point. The scale of perturbations are controlled by two parameters: $r$, the percentage of elements to be perturbed; $\epsilon$, the magnitude of the perturbation on each element. For example, a $r = 0.03$, $\epsilon = 0.5$ perturbation on $M$ represents that $3\%$ randomly selected elements of $M$ will be increased by $0.5 * |M|_\infty$, where $|M|_\infty$ denotes the largest element of $M$. The communication effectiveness under perturbation is measured when one agent's common ground has varied. Results on square matrices with perturbations on shared $M$ are presented here. Simulations on priors and rectangular matrices exhibit similar behaviors, see plots in Supp. Text Sec. D.

**Scaling Perturbation Size.** We investigate effectiveness under increasing perturbation. Matrices of size $100 \times 100$ are sampled as described above. Fixing $r = 0.03$, $\epsilon$ is altered as in $[0, 0.2, 0.4, 0.6, 0.8, 1]$. As shown in Fig. 1a, effectiveness drops for the one-step approximation comparing to Sinkhorn plans when the magnitude of perturbation increases, illustrating robustness of EOT to violations of common ground.

**Varying Matrix Dimension.** Fig. 1d shows the effects of matrix dimension. We fix $r = 0.03$, $\epsilon = 1$ and consider the dimension of $M$ in $[25, 50, 100, 200, 400]$. The probability that SK plans has higher

CI than its one-step approximation increases with the dimension of $M$. Moreover, the advantage of Sinkhorn planning is an effect that is increased in the presence of perturbations.

Fig. 1e. plots the average communication effectiveness for SK Plans and its one-step approximation with and without perturbations. Since the communication problem naturally gets harder as the dimension of M increases, we use the ratio between CI and the dimensional baseline to measure the communication effectiveness, in stead of CI.[6] Fig. 1e. suggests that communication effectiveness is more stable for SK plans under perturbations. Fig. 1f. plots the average difference in $L^1$-distance of the teaching (and learning) plan before and after perturbations. For instance, given $M$, denote the matrix after perturbation by $M_p$. Let $T^{sk}$, $T_p^{sk}$ be the teacher's SK plans obtained from EOT on $M$ and $M_p$ respectively. Their difference is measured as $|T^{sk} - T_p^{sk}|_{L_1}$. Fig. 1f. shows that under perturbation, the deviations on SK plans are considerably smaller than its one-step approximations.

## 4.2 Greedy selection of data

We investigate the effect of greedy parameter $\lambda$ on EOT when deviation occurs on agents' common ground. Fig. 1b-c plot the average CI of Sinkhorn planning for $50 \times 50$ matrices as $\lambda$ varies $[0.1, 0.5, 1, 5, 10, 20, 40]$. Fixing $r = 0.3$, $\epsilon = 0.3$, the hyperparameter $\alpha$ of Dirichlet distribution for sampling $M$ is set to be 10 in Fig. 1b, and 1 in Fig. 1c. ($\alpha$ for $P_0(\mathcal{H})$ is set to be 10 in both). The gap between the two curves expands in both Fig. 1b-c, which illustrates that the robustness of EOT decreases as $\lambda$ grows. As shown in Proposition 8, agents' optimal plan mainly concentrated on leading diagonals of their initial matrices. When deviation on $M$ causes mismatching leading diagonals for agents, $\lambda > 1$ exaggerates the difference, hence the drop on the CI. Notice that the rate of reduction of CI is more severe in 1b than 1c as $\lambda$ increases. This is consistent with the model prediction (Section 3.3) that under the same scale of perturbations, agents' plans are more likely to have variation on leading diagonals when element of the initial matrices are closer to evenly distributed.

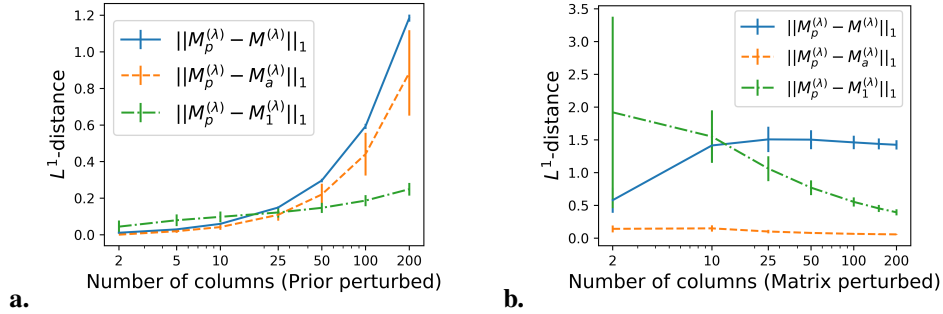

**a.**          **b.**

Figure 2: Mean and stdev of the $L_1$-distance between SK plan $M_p^{(\lambda)}$ of $M_p$ and its three estimations: original SK plan $M^{(\lambda)}$ (blue), linear approximation $M_a^{(\lambda)}$ (orange), and one step approximation $M_1^{(\lambda)}$ (green).

## 4.3 Linear approximation

The gradient guaranteed by Theorem 5 allows online correction of deviations in common ground via linear approximation. Let $M_p$ be a deviation of $M$ obtained by perturbing elements of $M$. To estimate the SK plan $(M_p^{(\lambda)})$ of $M_p$, we benchmark this linear approximation $M_a^{(\lambda)} = M^{(\lambda)} + \nabla_{\mathbf{r},\mathbf{c}} \Phi \cdot \delta(\mathbf{r}, \mathbf{c}) + \nabla_M \Phi \cdot \delta M$ against the original SK plan $M^{(\lambda)}$, and the one-step approximation $M_1^{(\lambda)}$ of $M_p^{(\lambda)}$ [7]. We use $L_1$-distance from each approximation to $M_p^{(\lambda)}$ to measure the error.

Fig. 2 shows the Monte-Carlo result of $10^5$ samples. $\lambda = 1$, $\mathbf{r}$ and $\mathbf{c}$ are uniform, and fix the number of rows to be 50. Matrices, which differ in the number of columns (labeled on x-axes, varying from 2

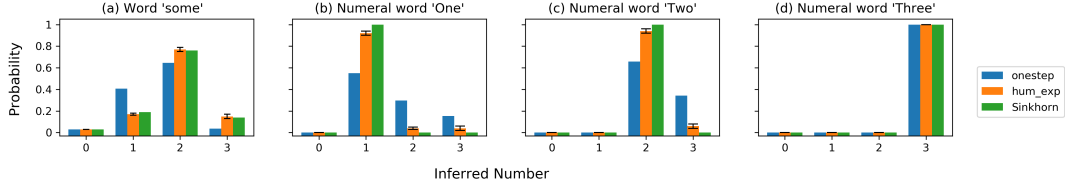

Figure 3: Green and blue bars plot the models' predictions regarding the learner's inference about the actual number of red apple based on the teacher's statement. The orange bars plot the empirical mean wager by the learner on each word state from Goodman and Stuhlmüller [2013].

to 200), are sampled so that each column follows Dirichlet distribution with parameter $\alpha = 1$. The perturbation on marginals are taken by adding $10\%$ to the sum of the first row while subtracting the same value from the sum of the second row (Fig. 2.a). The perturbation on matrices is the same as in Sec. 4.1 with $r = 0.03$ and $\epsilon = 0.5$ (Fig. 2.b). Linear approximation shows a modest effect for perturbations on the marginals, but is remarkably effective for perturbations on the matrix $M$.

### 4.4 An application to human data

We explore the following scenario from Goodman and Stuhlmüller [2013]. Three apples, which could be red or green, are on a table. The teacher looks at the table and make a statement quantifying the number of red apples such as "Some of the apples are red". The learner then infers the number of red apples based on the teacher's statement. The hypothesis set $\mathcal{H} = \{\text{'0','1','2','3'}\}$ represents the true number of red apples, and the data space $\mathcal{D} = \{\text{none, some, all}\}$ contains all the relevant quantifier words the teacher may choose. Hence, the shared (unnormalized) consistency matrix for

$$\text{both agents is } M = \begin{array}{c} \text{none} \\ \text{some} \\ \text{all} \end{array} \begin{pmatrix} \overset{\text{'0'}}{1} & \overset{\text{'1'}}{0} & \overset{\text{'2'}}{0} & \overset{\text{'3'}}{0} \\ 0 & 1 & 1 & 1 \\ 0 & 0 & 0 & 1 \end{pmatrix}. \text{ Both agents may estimate each other's likelihood}$$

matrix by normalizing $M$. The data were fit with a binomial prior distribution. Parameters for the one-step approximation as [Goodman and Stuhlmüller, 2013] were base rate 0.62, and $\lambda = 3.4$ and for EOT were base rate 0.82 (any choice of $\lambda$). Fig. 3(a) plots both models' predictions (i.e. learning plan) and the mean wager on the actual number of red apple by experimental participants, based on the teacher's statement [8]. In this case, both models successfully capture that 'some' implies 'not all'.

We further compare EOT and its one step approximation on interpretation of numerals. The setting is the same as above, except after looking at the table, the teacher makes a numeric statement such as "Two of the apples are red". Fig. 3(b-d) shows simulation results with priors over $\mathcal{H}$ and $\mathcal{D}$ be uniform and $\lambda = 1$. Notice that the EOT plan is in fact the identity matrix $I_4$. It is both more consistent with the human behavior experiments, and achieves the highest possible communicate effectiveness as $\text{CI}(I_4, I_4) = 1$, whereas the one-step approximation only has $\text{CI} = 0.5$.

## 5 Conclusions

Formalizing cooperative communication as Entropy regularized Optimal Transport, we show that cooperative communication is provably effective in terms of maximizing likelihood of belief transmission and is robust and adaptable to violations of common ground, with probabilistic reasoning optimizing the trade-off between effective belief transmission and robustness to deviations in common ground. Thus, claims regarding cooperative communication of beliefs between quite different agents, such as parents and children, speakers and listeners, teachers and learners, across cultures, or even between humans and machines, are mathematically well-founded. Our approach, based on unifying probabilistic and information theoretic models under Entropy regularized Optimal Transport, may lead to new formal foundations for theories of human-human and human-machine cooperation.

## Broader Impact

The theoretical approach introduced in this paper unifies models that have been proposed in the literatures on human language, education, and human-robot interaction—domains with significant societal implications. Our analysis highlights conditions under which they may be robust to violations of assumptions, and through mathematical analysis of previously algorithmic proposals, provides a means by which we may understand and improve the robustness of these models. This provides a mathematical framework within which we may understand their safe and responsible use in applications. More generally, the field of machine learning has not traditionally considered possibility that humans are a collaborative partner both in generating the datasets of interest and in using model's predictions. The theory advanced herein is explicitly models this collaboration toward the goal of more effective human-machine teaming. Thus, while the contributions of the current work are primarily theoretical, there are potential positive implications in areas of society interest.

## Acknowledgments and Disclosure of Funding

This project was supported by DARPA grant HR00112020039 the content of the information does not necessarily reflect the position or the policy of the Government, and no official endorsement should be inferred.

This material is based on research sponsored by the Air Force Research Laboratory and DARPA under agreement number FA8750-17-2-0146 and the Army Research Office and DARPA under agreement HR00112020039. The U.S. Government is authorized to reproduce and distribute reprints for Governmental purposes notwithstanding any copyright notation thereon.

This work was also supported by DoD grant 72531RTREP, NSF SMA-1640816, NSF MRI 1828528 to PS.

## Footnotes

[1] A general definition can be made for any pair of probability measures.

[2] Data, $d_i$, are consistent with a hypothesis, $h_j$, when $M_{ij} > 0$.

[3]All proofs are included in Section E of Supplementary Text (**ST**).

[4]General result on non-negative shared distributions is stated and proved in Supp.Text Section E.1

[5]The hyperparameter is set to be $0.1$ as sparse matrices are in general more sensitive to perturbations.

[6]The dimensional baseline for a $N \times N$ matrix $M$ is set to be $1/N$, which is the probability that the learner infers the hypothesis teacher has in mind without communication.

[7]Thus, $M_1^{(\lambda)}$ is obtained from $M_p$ by one step Sinkhorn scaling.

[8] Human data are measured based on Fig.2 of Goodman and Stuhlmüller [2013].

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
