[Supplementary Material]

# Supplementary Text of A mathematical theory of cooperative communication

**Pei Wang**
Rutgers University–Newark
peiwang@rutgers.edu

**Junqi Wang**
Rutgers University–Newark
junqi.wang@rutgers.edu

**Pushpi Paranamana**
Rutgers University–Newark
pushpi.paranamana@nd.edu

**Patrick Shafto**
Rutgers University–Newark
patrick.shafto@gmail.com

## A  Properties of Optimal Transport and Sinkhorn scaling

**Example A.1.  An application of Sinkhorn Scaling and Proposition 1.**

Let $\mathbf{r} = \mathbf{c} = (\frac{3}{8}, \frac{5}{8})$, and the cost matrix be $C = \begin{pmatrix} \log 1 & \frac{1}{3}\log 2 \\ \frac{2}{3}\log 2 & \log 1 \end{pmatrix}$. For $\lambda = 3$, we may obtain $P^{(3)}$ by applying SK scaling on $P^{[3]} = \begin{pmatrix} e^{-3\log 1} & e^{-3\cdot\frac{1}{3}\log 2} \\ e^{-3\cdot\frac{2}{3}\log 2} & e^{-3\log 1} \end{pmatrix} = \begin{pmatrix} 1 & 1/2 \\ 1/4 & 1 \end{pmatrix}$, which proceeds as follows: (a) row normalizing $P^{[3]}$ such that each row sum equals 1, giving $\begin{pmatrix} 2/3 & 1/3 \\ 1/5 & 4/5 \end{pmatrix}$; (b) multiplying the first row by 3/8 and second row by 5/8 giving $L_0 = \begin{pmatrix} 1/4 & 1/8 \\ 1/8 & 1/2 \end{pmatrix}$. Then similarly, column normalization of $L_0$ with respect to $\mathbf{c}$ outputs $T_1 = \begin{pmatrix} 1/4 & 1/8 \\ 1/8 & 1/2 \end{pmatrix}$. As $L_0 = T_1$, the SK scaling has converged with $P^{(3)} = T_1$. In general, multiple iterations may be required to reach the limit.

We now summarize some of important features about OT and SK.

Numerous results on SK iteration have been proved. For instance, assuming uniform marginal distributions, SK iteration of a square $M$ converges if and only if $M$ has at least one positive diagonal [Sinkhorn and Knopp, 1967] and the limit must be a doubly stochastic matrix, which can be written as a convex combination of permutation matrices [Dufossé and Uçar, 2016]. SK iteration can be viewed as a continuous map [Sinkhorn, 1972]. For positive matrices, we illustrate, this map is in fact smooth, in particular differentiable. This allows to show that the unifying OT framework is robust to various perturbations on the common grounds and to derive precise gradient formula to recover (linear approximate) optimal communication plans (Section 3.2).

After Sinkhorn and Knopp [1967], the convergence results regarding Sinkhorn scaling has further developed in various fields (see survey [Idel, 2016]). SK converges at a speed that is several orders of magnitude faster than other transport solvers [Cuturi, 2013, Allen-Zhu et al., 2017]. Sinkhorn plans have been extensively applied in machine learning algorithms, for example in barycenter estimation [Altschuler et al., 2017], supervised learning [Frogner et al., 2015], domain adaptation [Courty et al., 2017] and training GANs [Arjovsky et al., 2017].

There is a strong geometric intuition that underlies SK scaling via the cross-product ratio (Definition 7). Matrices converge to the same limit under SK scaling if and only if they are cross-ratio equivalent [Wang et al., 2019]. The space $\mathcal{K}(M)$ formed by all matrices with the same cross-product ratios as $M$ is a special manifold Fienberg [1968]. SK scaling moves $M$ along a path in $\mathcal{K}$ to $M^*$ — the unique intersection between $\mathcal{K}$ and the manifold determined by the linear marginal conditions [Fienberg et al., 1970].

Preservation of cross-product ratios over SK scaling implies that Sinkhorn Plans of EOT are invariant under cost matrices constructed for agents with different depths of SK. For instance in the illustration of Proposition 2 of the main text, instead of being naive, a learner could also be pragmatic who would reason about his estimation of the teacher's reasoning and interpret data accordingly using Bayes' rule, i.e. proportional to elements of $L_1$ which is row normalization of $T_0$. Denote the teacher's cost matrices based on $L_0$ and $L_1$ by $C_0^T$ and $C_1^T$ respectively. Because both $L_0$ and $L_1$ are derived from $M$ by applying Sinkhorn iteration, they are cross-ratio equivalent. So they have the same SK limit, i.e. Sinkhorn plans with respect to both $C_0^T$ and $C_1^T$ are the same. Thus, even though the teacher's estimation of the learner was not accurate, the teacher's plan is still optimal. Indeed, optimal teaching plans are equivalent for any learning matrix that is cross-ratio equivalent to the common ground $M$.

Strengthened by the rich theory of OT, our framework can be used to solve much broader questions. For example, general existence of OT planning between two arbitrary probability measures over any probability spaces are well-studied [Villani, 2008]. This provides us machinery to study cooperative communications between agents even when $\mathcal{H}$ and $\mathcal{D}$ are continuous spaces. Further existence of optimal communicative plans are guaranteed as general existence of optimal couplings. Moreover, OT plannings enjoy many other desirable features such as: the optimality passes to subsets, convexity of OT distance, which enables broader perspectives on approximate inference and computation of optimal plans.

## B    Unifying existing theories of cooperative communication

Existing models of cooperative communication can be unified as approximate inference for EOT. In this section, we demonstrate this point by expressing representatives of three broad classes of models as EOT.

### B.1    Full recursive reasoning is EOT.

Cooperative models that build on the classic Theory of Mind recursion are methods utilizing fully Bayesian inference. For instance, *cooperative inference* [Yang et al., 2018, Wang et al., 2019] and *pedagogical reasoning* [Shafto and Goodman, 2008, Shafto et al., 2014, 2012]. To simplify exposition, we will focus on the theory of cooperative inference and illustrate how Bayesian inference models fit into our unifying EOT framework.

The core of cooperative inference between two agents is that the teacher's selection of data depends on what the learner is likely to infer and vice versa. Let $P_{L_0}(h)$ be the learner's prior of hypothesis $h \in \mathcal{H}$, $P_{T_0}(d)$ be the teacher's prior of selecting data $d \in \mathcal{D}$, $P_T(d|h)$ be the teacher's posterior of selecting $d$ to convey $h$ and $P_L(h|d)$ be the learner's posterior for $h$ given $d$. **Cooperative inference** emphasizes that agents' optimal conditional communication plans, $T^\star = P_T(\mathcal{D}|\mathcal{H})$ and $L^\star = P_L(\mathcal{H}|\mathcal{D})$ should satisfy the following system of interrelated equations for any $d \in \mathcal{D}$ and $h \in \mathcal{H}$, where $P_L(d)$ and $P_T(h)$ are the normalizing constants:

$$P_L(h|d) = \frac{P_T(d|h)\,P_{L_0}(h)}{P_L(d)} \qquad P_T(d|h) = \frac{P_L(h|d)\,P_{T_0}(d)}{P_T(h)} \tag{1}$$

Results in Yang et al. [2018] indicates that assuming uniform priors on $\mathcal{D}$ and $\mathcal{H}$, Eq.(1) can be solved using Sinkhorn iteration on the joint distribution $M$. More generally, we show:

**Proposition 3.** *Optimal conditional communication plans, $T^\star$ and $L^\star$, of a cooperative inference problem with arbitrary priors, can be obtained through Sinkhorn scaling. In particular, as a direct consequence, cooperative inference is a special case of the unifying EOT framework with $\lambda = 1$.*

### B.2    One-step approximate inference

Models in social cognitive development and pragmatic reasoning, including *Naive Utility Calculus* [Jara-Ettinger et al., 2016, Jern et al., 2017], *Rational Speech Act* (RSA) theory [Goodman and Stuhlmüller, 2013, Goodman and Frank, 2016, Franke and Jäger, 2016] and *Bayesian Teaching* [Eaves Jr and Shafto, 2016, Eaves Jr et al., 2016] and their extensions [Jara-Ettinger et al., 2015b, Baker et al., 2017, Jara-Ettinger et al., 2015a, Liu et al., 2017, Hamlin et al., 2013, Jara-Ettinger et al., 2015c, Bridgers et al., 2016, Gweon and Asaba, 2018, Gweon et al., 2014, Jara-Ettinger et al.,

2017, Cohn-Gordon et al., 2018, Ong et al., 2015, 2019] approximate cooperation as a single step of recursion.

For instance, RSA models the communication between a speaker and a listener, formalizing cooperation that underpins pragmatic language. A pragmatic speaker selects an utterance optimally to inform a naive listener about a world state. Whereas a pragmatic listener interprets an utterance rationally and infers the state using one step Bayesian inference. This represents a communicative process where a speaker-listener pair can be viewed as a teacher-learner pair with world states-utterances being hypotheses-data points, respectively.

RSA distinguishes among three levels of inference: a *naive listener*, a *pragmatic speaker* and a *pragmatic listener* [Goodman and Stuhlmüller, 2013]. A naive listener interprets an utterance according to its literal meaning. That is, given a shared matrix $M$, the naive listener's probability of selecting $h_i$ given $d_j$ is the $ij$-th element of $L_0$, which is obtained by row normalization of $M$.

A pragmatic speaker selects an utterance to convey the state such that maximizes utility. In particular, they pick $d_i$ to convey $h_j$ by soft-max optimizing expected utility,

$$P_{\mathrm{T}}(d_i|h_j) \propto e^{\alpha\,U(d_i;h_j)}, \tag{2}$$

where utility is given by $U(d_i; h_j) = \log L_0(h_j|d_i) - S(d_i)$, which minimizes the surprisal of a naive listener when inferring $h_j$ given $d_i$ with an utterance cost $S(d_i)$. This formulation is the same as one step of SK iteration in EOT framework (see Eq.(2) and Eq.(3)) where $C^T = -U(d;h)$, $\lambda = -\alpha$.

Next, a pragmatic listener reasons about the pragmatic speaker and infers the hypothesis using Bayes rule,

$$P_L(h_j|d_i) \propto P_T(d_i|h_j)P_L(h_j), \tag{3}$$

Here $P_{\mathrm{T}}(d_i|h_j)$ represents the listener's reasoning on the speaker's data selection and $P_{\mathrm{L}}(h_j)$ is the learner's prior. This is again one step recursion of EOT framework of $\lambda = 1$.

As described above, teaching and learning plans in RSA are one-step approximations of the Sinkhorn plans. EOT framework suggests that in many cases, such approximations are far from optimal. For example, world states are often referred at many levels of specificity by human agents [Graf et al., 2016, Hawkins et al., 2018], which yield a *upper triangular* joint distribution matrix. EOT would output a *diagonal matrix* as optimal plan which achieves the highest communication effectiveness, whereas cooperative index of one step approximation is much lower. Furthermore, one-step approximation plans are much more sensitive to agents' estimation of the other agent. For instance, a pragmatic speaker's teaching plan is tailored for a naive listener, in contrast the optimal plan obtained through fully recursion is stable for any listener derived from the same common ground.

### B.3 Single-step argmax approximation

Many recent advances in robotics involve artificial agents that implement human-like inverse planning [Fisac et al., 2017, Jara-Ettinger, 2019], such as simple or structured desire inference [Baker et al., 2009, Velez-Ginorio et al., 2017, Reddy et al., 2018], path and motion planning Kim and Pineau [2016], Dragan et al. [2013], pedagogical interaction [Ho et al., 2016, 2018] and value alignment Hadfield-Menell et al. [2016], Milli et al. [2017]. In cooperative inverse reinforcement learning, instead of selecting acts probabilistically, the maximum probability action is selected. For example, [Fisac et al., 2017] introduces *Pragmatic-Pedagogic Value Alignment*, a framework that is grounded in empirically validated cognitive models related to pedagogical teaching and pragmatic learning.

Pragmatic-pedagogic value alignment formalizes the cooperation between a human and a robot who perform collaboratively with the goal of achieving the best possible outcome according to an objective. The true objective however is only known to the human. The human performs pedagogical actions to teach the true objective to the robot. After observing human's action, the robot, who is pragmatic, updates his beliefs and perform an action that maximizes expected utility. The human, observing this action, can then update their beliefs about the robot's current beliefs and choose a new pedagogic action. Denote actions by $d$ and objectives by $h$. We can see that when the human performs the action they act as a teacher and when robot is performing the action it is vice versa.

In particular, the pedagogic human selects an action $d_i$ to teach the objective $h_j$ according to Eq. (2), where $U$ is the utility that captures human's best expected outcome. As described in Section B.2, this is equivalent to a single step recursion in the EOT framework.

Denote the robot's prior belief distribution on the objectives by $P_R(h_j)$. The robot interprets the human's action $d_i$ rationally and updates his beliefs about the true objective using Bayes rule as Eq. (3). Then acting as a teacher, the robot chooses an action that maximizes the human's expected utility using argmax function:

$$P_R(d_i) = \arg\max_{d_R} \sum_{d_H, h_j} U(d_R, d_H; h) \cdot P_R(h_j)$$

where, $d_R$ denotes the robot's actions and $d_H$ denotes the human's actions. Unlike in human communication [Eaves Jr and Shafto, 2016, Eaves Jr et al., 2016] where the plans are chosen proportionally to a probability distribution, here the robot chooses a deterministic action using argmax function.

As described above, inverse planning in robotics is modeled by computing a single step of Sinkhorn iteration and selecting the action that maximizes the outcome. Unlike full recursive reasoning is EOT, which tends to select the leading diagonal of the common ground $M$ as $\lambda \to \infty$ (Proposition 8), inverse planning methods like pragmatic-pedagogic value alignment selects the maximal element in each column of $M$, which is not even guaranteed to form a plan to distinguish every hypothesis. Hence a big concern of such argmax method is that for large hypothesis spaces, multiple hypotheses may reach argmax on the same data which lead to low communication efficiency. Further, continuity is generally lost for deterministic methods as argmax, which reduces the models' robustness comparing to EOT.

In summary, EOT framework unifies existing models of cooperative communication in social cognitive development, pragmatic reasoning and robotics with cooperative agents for specific missions and inference with different Sinkhorn iteration depths. This unification not only allows one to draw strong comparison of the relative merits and predictions of different theories, but also establish a potential toolbox for one to design assignment tailored models, which could achieve the best balance between efficiency and accuracy.

## C   Further discussion on Sensitivity for large $\lambda$

Sensitivity to perturbations is a concern as $\lambda \to \infty$. Figure 1 demonstrates an example where a slight variation on the initial matrices $M_1$ and $M_2$ can result a huge difference on $M_1^{(\lambda)}$ and $M_2^{(\lambda)}$ as $\lambda$ approaches infinity. The figure plots the Sinkhorn plans derived from $M_i^{(\lambda)}$ with the starting matrices $M_1$, $M_2$ differing from $M$ only by 2% on their $l^\infty$-distance. However, in this particular case, the change makes a huge difference: $M$ has two leading diagonals, while the perturbed $M_1$ and $M_2$ of $M$ enhanced one for each, making each $M_1$ and $M_2$ has only one leading diagonal. When $\lambda$ approaches zero, all products of diagonals tends to be the same, thus the curves (red for $M$, green for $M_1$ and blue for $M_2$) converges to a common limit point, the uniform matrix. But as $\lambda$ increases, the leading diagonals overwhelm other diagonals, and results in a fixed divergence on the limit when $\lambda \to \infty$. Therefore, in this case, no matter how slight the changes are, as long as they modify the set of leading diagonals, there will be a fixed difference on the limits when $\lambda \to \infty$ according to the leading diagonals. Thus, $M^{(\infty)}$ is no longer continuous on the initial matrix $M$.

In particular, as $\lambda$ increases, the cooperative index, $\text{CI}(M_1^{(\lambda)}, M_2^{(\lambda)})$, between two agents with initial matrix $M_1$ and $M_2$ will be very small, even zero, if there is no overlapping positive element between $M_1^{(\lambda)}$ and $M_2^{(\lambda)}$ whereas $\text{CI}(M_1^{(1)}, M_2^{(1)})$ is bounded from below by the reciprocal of the number of diagonals of $M$.

**Example C.1.** Assume that the teacher has the accurate $M = \begin{pmatrix} 1 & 5 & 0 \\ 0 & 1 & 6 \\ 0 & 0 & 1 \end{pmatrix}$. For any $\lambda$, the optimal teaching plan $T^{(\lambda)} = I_3$. Suppose the learner gets constant noise of size 0.1 in the position of $M_{31}$. When $\lambda = 1$, the learner's initial matrix is $L^{[\lambda=1]} = \begin{pmatrix} 1 & 5 & 0 \\ 0 & 1 & 6 \\ 0.1 & 0 & 1 \end{pmatrix}$, the corresponding optimal plan is $L^{(\lambda=1)} = \begin{pmatrix} 0.41 & 0.51 & 0 \\ 0 & 0.41 & 0.51 \\ 0.51 & 0 & 0.41 \end{pmatrix}$ and $\text{CI}(T^{(\lambda)}, L^{(1)}) = 0.41$. Similarly when $\lambda = 2$, we have $L^{[\lambda=2]} = \begin{pmatrix} 1 & 25 & 0 \\ 0 & 1 & 36 \\ 0.01 & 0 & 1 \end{pmatrix}$, $L^{(\lambda=2)} = \begin{pmatrix} 0.25 & 0.75 & 0 \\ 0 & 0.25 & 0.75 \\ 0.75 & 0 & 0.25 \end{pmatrix}$ and $\text{CI}(T^{(\lambda)}, L^{(2)}) = 0.25$. Furthermore,

Figure 1: Lost of Continuity when $\lambda \to \infty$

as $\lambda \to \infty$, $L^{(\lambda)} \to \begin{pmatrix} 0 & 1 & 0 \\ 0 & 0 & 1 \\ 1 & 0 & 0 \end{pmatrix}$ and $\mathrm{CI}(T^{(\lambda)}, L^{(\lambda)}) \to 0$ . Thus, in this case communication efficiency is completely vanished due to deviations between the teacher and learner are exaggerated by greedy selection of examples.

# D  Simulations

## D.1  Perturbation on common ground and Greedy selection of data

Figure 2: **a.** The Cooperative Index (CI) of Sinkhorn plans (SK) and its one step approximation (onestep) as total perturbation increases. **b-d.** $r = 0.03$, $\epsilon = 1$, dimension of $M$ varies as shown in $x$-axis. **b.** The probability of SK has higher CI than onestep. **c.** The average communication effectiveness for SK and onestep with and without perturbations denoted by SK-p, onestep-p, SK-np, onestep-np accordingly. **d.** The average difference of the teaching (and learning) plan for SK (and one-step approximation) before and after perturbations, measured by the $L^1$-distance.

**Rectangular matrices.** Figure 2 are plots based on stimulation of matrix perturbation on rectangular matrices. The number of columns for sampled matrices is fixed to be 50. The number of rows varies as in $[10, 25, 50, 100, 150, 200]$. All the other parameters are the same as in the main text: $r = 0.03$, $\epsilon = 1$ and parameter of Dirichlet distribution is $0.1$ for both initial matrix $M$ and prior over $\mathcal{H}$.

**Prior perturbation.** Figure 3 are plots based on stimulation of prior perturbation on square matrices. In **(a-c)**, the matrix size varies as in $[25, 50, 100, 200, 400]$, parameter of Dirichlet distribution is $0.1$

for both initial matrix $M$ and prior over $\mathcal{H}$. We increase the perturbation rate to $r = 0.07$ and reduce the magnitude to $\epsilon = 0.15$ as the prior over $\mathcal{H}$ contains considerably fewer number of elements than $M$. In **(d)**, the matrix size is fixed to be $50 \times 50$, parameter of Dirichlet distribution for initial matrix is 10, for prior is 1, $r = 0.3$ and $\epsilon = 0.3$. In general, we observer that both Sinkhorn plans and its one step approximation are much more sensitive to matrix perturbations than prior perturbations.

Figure 3: **a-c.** $r = 0.07$, $\epsilon = 0.15$, dimension of $M$ varies as shown in $x$-axis. **a.** The probability of SK has higher CI than onestep. **b.** The average communication efficiency for SK and onestep with and without perturbations denoted by SK-p, onestep-p, SK-np, onestep-np accordingly. **c.** The average difference of the learning plan for SK and one-step approximation before and after perturbations, measured by the $L^1$-distance. **d.** The average CI for $50 \times 50$ matrices as $\lambda$ varies in $[0.1, 0.5, 1, 5, 10, 20]$.

## D.2 Linear Approximations

Figure 4 shows the result of comparisons on different approximations of Sinkhorn limits of perturbed matrices/marginals, with different choices of Dirichlet hyperparameter $\alpha = 0.1, 10$ ($\alpha = 1$ in the main paper). Other parameters (matrix size, sample size and method, and perturbation patterns) are the same as in the main text.

# E   Proofs of Propositions

**Proposition 3.** *Optimal conditional communication plans, $T^\star$ and $L^\star$, of a cooperative inference with arbitrary priors denoted by $P_{T_0}(\mathcal{D})$ and $P_{L_0}(\mathcal{H})$, can be obtained through Sinkhorn scaling. In particular, as a direct consequence, cooperative inference is a special case of the unifying EOT framework with $\lambda = 1$.*

*Proof.* Consider cooperative inference as in Eq. (5) of the main content, we may rewrite it as follows:

$$
\begin{aligned}
P_L(h|d)P_{T_0}(d) &= \frac{P_T(d|h)P_{L_0}(h)P_{T_0}(d)}{P_L(d)} \\
P_T(d|h)P_{L_0}(h) &= \frac{P_L(h|d)P_{T_0}(d)P_{L_0}(h)}{P_T(h)}
\end{aligned}
\tag{4}
$$

Figure 4: **a.** Perturb on row sums, with $\alpha = 0.1$; **b.** Perturb on row sums, with $\alpha = 10$; **c.** Perturb on matrices, with $\alpha = 0.1$; **d.** Perturb on matrices, with $\alpha = 10$.

which is equivalent to

$$P_L(h|d)P_{T_0}(d) = \frac{P_T(d|h)P_{L_0}(h)}{P_L(d)/P_{T_0}(d)}, \tag{5a}$$

$$P_T(d|h)P_{L_0}(h) = \frac{P_L(h|d)P_{T_0}(d)}{P_T(h)/P_{L_0}(h)}. \tag{5b}$$

Notice that Eq. (5) is the stable condition of Sinkhorn scaling on $\widetilde{M} = P_L(h|d)P_{T_0}(d)$ with $\mathbf{r} = P_{T_0}(\mathcal{D})$, $\mathbf{c} = P_{L_0}(\mathcal{H})$. Hence Eq. (5) can be solved using fixed-point iteration as explored in [Shafto et al., 2014]: for the first evaluation of the left hand side of (5a), initialize $P_L(h|d)$ by $P_{L_0}(h|d)$ which is the row normalization of the shared distribution $M = P(d,h)$ and denote $P_{L_0}(h|d) \cdot P_{T_0}(d)$ by $\widetilde{L}_0$. Then the first evaluation of the left hand side of (5b), denoted by $\widetilde{T}_1$, can be obtained by column normalizing $\widetilde{L}_0$ with respect to $\mathbf{c}$. Next, the second evaluation of (5a) is achieved by row normalizing of $\widetilde{T}_1$ with respect to $\mathbf{r}$, and iterate this process until convergence. This is precisely $(\mathbf{r}, \mathbf{c})$-Sinkhorn scaling starting with $\widetilde{L}_0$. Symmetrically, (5) can also be solved by $(\mathbf{r}, \mathbf{c})$-Sinkhorn scaling starting with $\widetilde{T}_0 = P_{T_0}(d|h) \cdot P_{L_0}(h)$.

Let $M$ be the shared distribution, $\mathbf{r} = P_{T_0}(\mathcal{D})$ be the teacher's prior and $\mathbf{c} = P_{L_0}(\mathcal{H})$ be learner's prior. As shown in the above paragraph, after cooperative inference, the teacher's conditional communication plan $T^\star$ is the limit of $(\mathbf{c}, \mathbf{r})$-SK scaling of $\widetilde{L}_0 = (P_{L_0}(h_j|d_i)P_{T_0}(d_i))$. On the other hand, under the unifying EOT framework, the optimal teaching plan $T^{(\lambda=1)}$ is the limit of $(\mathbf{c}, \mathbf{r})$-SK scaling of $\widehat{L}_0 = (P_{L_0}(h_j|d_i)e^{S_T(d_i)})$ based on Eq. (4). When the teacher's expense $S_T(d_i)$ of selecting $d_i$ is proportional to $\log P_{T_0}(d_i)$, $T^{(1)} = T^\star$. Symmetrically, one may check the same holds for $L^{(1)} = L^\star$.

$\square$

**Proposition 8.** *Assuming uniform marginals, $M^{(\lambda)}$ is concentrating around the leading diagonals of $M$ as $\lambda \to \infty$.*

*Proof.* Let $D_\sigma, D_{\sigma'}$ be two diagonals of a $n \times n$ shared matrix $M$ and $d_\sigma, d_{\sigma'}$ be products of their elements respectively (Definition 6). Further, let the diagonals in $M^{[\lambda]}$ determined by the same $\sigma$ and $\sigma'$ be $D_\sigma^{[\lambda]}$ and $D_{\sigma'}^{[\lambda]}$. Their cross product ratio is denoted by $\mathrm{CR}(D_\sigma^{[\lambda]}, D_{\sigma'}^{[\lambda]})$. If $D_{\sigma'}$ is a leading

diagonal and $D_\sigma$ is not, then $d_\sigma/d_{\sigma'} < 1$, and so $\mathrm{CR}(D_\sigma^{[\lambda]}, D_{\sigma'}^{[\lambda]}) = (d_\sigma/d_{\sigma'})^\lambda \to 0$ as $\lambda \to \infty$ (Fact $A$). If both $D_\sigma$ and $D_{\sigma'}$ are leading diagonals, then $d_\sigma/d_{\sigma'} = 1$, and so $\mathrm{CR}(D_\sigma^{[\lambda]}, D_{\sigma'}^{[\lambda]}) = (d_\sigma/d_{\sigma'})^\lambda \to 1$ as $\lambda \to \infty$. We now show that for any element $M_{st}^{(\lambda)}$ of $M^{(\lambda)}$, if the corresponding element $M_{st}$ is not on a leading diagonal of $M$, then $M_{st}^{(\lambda)} \to 0$. It is clear that if $M_{st}$ is not contained in any positive diagonal of $M$, then $M_{st}^{(\lambda)} \to 0$ as off diagonal elements vanishes along Sinkhorn iteration [Wang et al., 2019]. Now suppose that $M_{st}$ is contained in a non-leading positive diagonal determined by permutation $\sigma$. If $M_{st}^{(\lambda)}$ does not vanish, there exists an $\epsilon > 0$ such that $M_{st}^{(\lambda)} > \epsilon$ for any $\lambda$. And so $M_{st}^{(\lambda)}$ must be contained in a positive diagonal of $M^{(\lambda)}$. Without loss, we may assume $M_{st}^{(\lambda)}$ is the smallest non-vanishing element that is off leading diagonals of $M$. Then $d_\sigma^{(\lambda)} > \epsilon^n$, and so $d_\sigma^{(\lambda)}/d_{\sigma'}^{(\lambda)} > \epsilon^n$ because $d_{\sigma'}^{(\lambda)} \leq 1$ ($M^{(\lambda)}$ is a joint distribution). This is contradiction to Fact $A$. Therefore, $M^{(\lambda)}$ is concentrating around the leading diagonals of $M$ as $\lambda \to \infty$. $\qquad\square$

Wang et al. [2019] explored the sensitivity of $\Phi$ to perturbation on elements in $M$. They showed that $\Phi$ is continuous on $M$. In particular, they demonstrated that $\Phi$ is robust to any amount of off-diagonal perturbations on $M$. SK scaling is also continuous on its scalars. Let $\mathbf{r}^\epsilon$ and $\mathbf{c}^\epsilon$ be vectors obtained by varying elements of $\mathbf{r}$ and $\mathbf{c}$ at most by $\epsilon$, where $\epsilon > 0$ quantifies the amount of perturbation. Distances between vectors or matrices are measured by $l^\infty$ norm (the maximum element-wise difference), e.g. $d(\mathbf{r}^\epsilon, \mathbf{r}) \leq \epsilon$. We prove that $\Phi$ is continuous on $\mathbf{r}$ and $\mathbf{c}$, thus the following holds:

**Proposition 4.** *For any joint distribution $M$ and positive marginals $\mathbf{r}$ and $\mathbf{c}$, if $\Phi(M, \mathbf{r}^\epsilon, \mathbf{c}^\epsilon)$ and $\Phi(M, \mathbf{r}, \mathbf{c})$ exist, then $\Phi(M, \mathbf{r}^\epsilon, \mathbf{c}^\epsilon) \to \Phi(M, \mathbf{r}, \mathbf{c})$ as $\mathbf{r}^\epsilon \to \mathbf{r}, \mathbf{c}^\epsilon \to \mathbf{c}$.*

*Proof.* Note that the continuity of $\Phi$ on the marginals is independent of the choice of a particular $\lambda$, we will drop the $\lambda$ for the rest of the proof to make the notation neater. Sinkhorn scaling of $M$ converges with marginal conditions $(\mathbf{r}, \mathbf{c})$ and $(\mathbf{r}^\epsilon, \mathbf{c}^\epsilon)$ implies that $\sum_{i=1}^n r_i = \sum_{j=1}^m c_j$ and $\sum_{i=1}^n r_i^\epsilon = \sum_{j=1}^m c_j^\epsilon$ (see Menon and Schneider [1969]). Let $k = \sum_{i=1}^n r_i$ and $k^\epsilon = \sum_{i=1}^n r_i^\epsilon$. We will prove in three steps. First, we show the claim when $k = k^\epsilon$. As $k = k^\epsilon$, at least two elements in $\mathbf{r}$ (or $\mathbf{c}$) are perturbed. Without loss, we will assume that only two elements, $r_s$ and $r_t$ in $\mathbf{r}$, are varied by amount $\epsilon$ since the general case may be treated as compositions of such. Then for $\mathbf{r}^\epsilon = (r_1^\epsilon, \dots, r_n^\epsilon)$, we have $r_s^\epsilon = r_s + \epsilon$, $r_t^\epsilon = r_t - \epsilon$ and $r_i^\epsilon = r_i$ if $i \neq s$ or $t$. Let $\Phi(M, \mathbf{r}, \mathbf{c}) = M^*$, $M^{*\epsilon}$ be the matrix obtained from varying the element $M_{s1}^*$ and $M_{t1}^*$ of $M^*$ by $\epsilon$ and $-\epsilon$, i.e. $M_{s1}^{*\epsilon} = M_{s1}^* + \epsilon$, $M_{t1}^{*\epsilon} = M_{t1}^* - \epsilon$ and $M_{ij}^{*\epsilon} = M_{ij}^*$ otherwise. Then the statement can be verified as following:

$$d(\Phi(M, \mathbf{r}, \mathbf{c}), \Phi(M, \mathbf{r}^\epsilon, \mathbf{c})) \overset{(a)}{=} d(M^*, \Phi(M^*, \mathbf{r}^\epsilon, \mathbf{c}))$$

$$\overset{(b)}{\leq} d(M^*, \Phi(M^{*\epsilon}, \mathbf{r}^\epsilon, \mathbf{c})) + d(\Phi(M^{*\epsilon}, \mathbf{r}^\epsilon, \mathbf{c}), \Phi(M^*, \mathbf{r}^\epsilon, \mathbf{c}))$$

$$\overset{(c)}{=} d(M^*, M^{*\epsilon}) + d(\Phi(M^{*\epsilon}, \mathbf{r}^\epsilon, \mathbf{c}), \Phi(M^*, \mathbf{r}^\epsilon, \mathbf{c}))$$

$$\overset{(d)}{=} \epsilon + d(\Phi(M^{*\epsilon}, \mathbf{r}^\epsilon, \mathbf{c}), \Phi(M^*, \mathbf{r}^\epsilon, \mathbf{c})) \overset{(e)}{\to} 0 \text{ as } \epsilon \to 0$$

where $(a)$ holds since $M^*$ and $M$ are cross-ratio equivalent and must converge to the same limit under any Sinkhorn scaling; $(b)$ is triangle inequality; $(c)$ holds since $M^{*\epsilon}$ is already $(\mathbf{r}^\epsilon, \mathbf{c})$-normalized, hence $\Phi(M^{*\epsilon}, \mathbf{r}^\epsilon, \mathbf{c}) = M^{*\epsilon}$; $(d)$ holds as $d(M^*, M^{*\epsilon}) = \epsilon$ by construction; $(e)$ holds because $\Phi$ is continuous on $M$ proved in Sinkhorn [1972].

Now we show the case where $k \neq k^\epsilon$, but the proportion between corresponding elements in $\mathbf{r}$ and $\mathbf{r}^\epsilon$ are the same, thus $r_i^\epsilon/r_i = r_j^\epsilon/r_j = \alpha$. Let $M^{*\alpha} = \alpha * M^*$, i.e. $M_{ij}^{*\alpha} = \alpha * M_{ij}^*$. Since $M^{*\alpha}$ is $(\mathbf{r}^\epsilon, \mathbf{c})$ normalized and also has the same cross ratios of $M$, $\Phi(M, \mathbf{r}^\epsilon, \mathbf{c}) = M^{*\alpha}$. Note that $d(M^{*\alpha}, M^*) \leq \epsilon$, so $\Phi(M, \mathbf{r}^\epsilon, \mathbf{c}) \to \Phi(M, \mathbf{r}, \mathbf{c})$ as $\epsilon \to 0$.

Finally for the general case, where $k \neq k^\epsilon$ and elements of $\mathbf{r}$ and $\mathbf{r}^\epsilon$ are not proportional. Let $\mathbf{r}^\alpha = (k^\epsilon/k) * \mathbf{r}$. Then elements of $\mathbf{r}$ and $\mathbf{r}^\alpha$ are proportional and $\sum r_i^\alpha = \sum r_i^\epsilon = k^\epsilon$. Thus based on the previous two cases, we have $d(\Phi(M, \mathbf{r}, \mathbf{c}), \Phi(M, \mathbf{r}^\epsilon, \mathbf{c})) \leq d(\Phi(M, \mathbf{r}, \mathbf{c}), \Phi(M, \mathbf{r}^\alpha, \mathbf{c})) + d(\Phi(M, \mathbf{r}^\alpha, \mathbf{c}), \Phi(M, \mathbf{r}^\epsilon, \mathbf{c})) \to 0$ as $\epsilon \to 0$. Hence, we are done. $\qquad\square$

### E.1 General version of Theorem 5

Enlightened by Luise et al. [2018], we can conclude a stronger version of the smoothness of $\Phi$ in the following way:

**Definition.** A pattern $\mathfrak{P}$ is a subset of $\{1, 2, \ldots, n\} \times \{1, 2, \ldots, m\}$, and a matrix $M = (M_{ij})$ of pattern $\mathfrak{P}$ is a non-negative matrix with $M_{ij} > 0$ if and only if $(i, j) \in \mathfrak{P}$. In this paper, $M$ is not allowed to have a vanishing row or column.

**Theorem 5** (General venison of Theorem 5). *Let $(\mathfrak{P}, \mathfrak{D})$ be a pair where $\mathfrak{P}$ is a pattern, and where $\mathfrak{D} \subseteq (\mathbb{R}^+)^n \times (\mathbb{R}^+)^m$ is the set consisting of vectors $(\mathbf{r}, \mathbf{c}) \in (\mathbb{R}^+)^n \times (\mathbb{R}^+)^m$ satisfying the equivalent conditions in Theorem 2 of Rothblum and Schneider [1989], in other words, pattern $\mathfrak{P}$ is exact $(\mathbf{r}, \mathbf{c})$-scalable. Let $\mathcal{M}_{\mathfrak{P}} = (\mathbb{R}^+)^{\mathfrak{P}}$ be the open cone of nonnegative matrices of pattern $\mathfrak{P}$, then for a given $\lambda \in (0, \infty)$, $\Phi : \mathcal{M}_{\mathfrak{P}} \times \mathfrak{D} \to \mathcal{M}_{\mathfrak{P}}$ is smooth.*

*Proof.* We use the same strategy as the proof of Theorem 2 in Luise et al. [2018]. Throughout the proof, let $\lambda \in (0, \infty)$ be a fixed positive real number.

First we make a decomposition of $\Phi$. This is possible because the exact scaling conditions guarantee the existence of diagonal matrices $D_1, D_2$ such that $\Phi(M, \mathbf{r}, \mathbf{c}) = M^{(\lambda)} = D_1 M^{[\lambda]} D_2$, equivalently, there exist a pair of vectors $(\alpha, \beta) \in \mathbb{R}^n \times \mathbb{R}^m$ such that $\Phi(M, \mathbf{r}, \mathbf{c}) = \mathrm{diag}(e^{\lambda \alpha}) M^{[\lambda]} \mathrm{diag}(e^{\lambda \beta})$. The pair $(D_1, D_2)$ is unique up to a scalar $d \in \mathbb{R}^+$ with actions $d : (D_1, D_2) \mapsto (dD_1, d^{-1}D_2)$, thus the pair of vectors $(\alpha, \beta)$ is unique up to a constant $\delta : (\alpha, \beta) \mapsto (\alpha + \delta, \beta - \delta)$ (plus/minus the same number on each element of the vectors). So we may always assume that the last component of $\beta$ vanishes, i.e., $\beta_m = 0$. In the following text, we use $\bar{\beta}$ to denote the first $m - 1$ components of $\beta$, and if $\bar{\beta}$ occurs, the corresponding $\beta$ is the vector by appending a 0 at the end of $\bar{\beta}$.

Then we can decompose the map $\Phi$ into the composition of two other maps: $\Phi = \mu \circ (\rho, \Psi)$. Here the map $\rho : \mathcal{M}_{\mathfrak{P}} \times \mathfrak{D} \to \mathcal{M}_{\mathfrak{P}}$ is the regularization map (regardless of the marginal conditions) $\rho(M, (\mathbf{r}, \mathbf{c})) = M^{[\lambda]}$, the map $\Psi : \mathcal{M}_{\mathfrak{P}} \times \mathfrak{D} \to \mathbb{R}^n \times \mathbb{R}^m$ maps $(M, \mathbf{r}, \mathbf{c})$ to the pair of vectors $(\alpha, \beta)$ with $\beta_m = 0$ as in the above discussion (such that $\Phi(M, \mathbf{r}, \mathbf{c}) = \mathrm{diag}(e^{\lambda \alpha}) M^{[\lambda]} \mathrm{diag}(e^{\lambda \beta})$), and the map $\mu : \mathcal{M}_{\mathfrak{P}} \times \mathbb{R}^n \times \mathbb{R}^m \to \mathcal{M}_{\mathfrak{P}}$ is such that $\mu(P, \alpha, \beta) = \mathrm{diag}(e^{\lambda \alpha})(P) \mathrm{diag}(e^{\lambda \beta})$. It can be easily seen that from the definitions the decomposition $\Phi = \mu \circ (\rho, \Psi)$ is valid.

Next, having this decomposition, we just need to show that $\mu, \rho$ and $\Psi$ are smooth, then $\Phi$ as the composition of smooth maps remains smooth.

(Smoothness of $\Psi$:) We use the same strategy as Theorem 2 in Luise et al. [2018]. Define the Lagrangian

$$\mathcal{L}(M, \mathbf{r}, \mathbf{c}; \alpha, \beta) = -\mathbf{r}^\top \alpha - \mathbf{c}^\top \beta + \sum_{(i,j) \in \mathfrak{P}} \frac{e^{\lambda \alpha_i} M_{ij}^\lambda e^{\lambda \beta_j}}{\lambda}.$$

where $\Psi(M, \mathbf{r}, \mathbf{c}) = (\alpha, \beta)$ optimizes $\mathcal{L}$ for fixed $M, \mathbf{r}, \mathbf{c}$ as proved in Luise et al. [2018], Cuturi [2013]. By smoothness of $\mathcal{L}$ (easy to see from expression), we may conclude that $N := \nabla_{(\alpha, \bar{\beta})} \mathcal{L}$ is $C^k$ for any $k \geq 0$ and $\nabla_{(\alpha, \bar{\beta})} \mathcal{L}(M, \mathbf{r}, \mathbf{c}; \Psi(M, \mathbf{r}, \mathbf{c})) = \mathbf{0}$ for any $M, \mathbf{r}, \mathbf{c}$.

Fix $(M_0, \mathbf{r}_0, \mathbf{c}_0; \alpha_0, \beta_0)$ such that $N(M_0, \mathbf{r}_0, \mathbf{c}_0; \alpha_0, \beta_0) = \mathbf{0}$ and $(\beta_0)_m = 0$. Since $\nabla_{(\alpha, \bar{\beta})} N = \nabla_{(\alpha, \bar{\beta})} \otimes \nabla_{(\alpha, \bar{\beta})} \mathcal{L}$ is the Hessian of the strictly convex function $\mathcal{L}$, then $\nabla_{(\alpha, \bar{\beta})} N(M_0, \mathbf{r}_0, \mathbf{c}_0; \alpha_0, \beta_0)$ is invertible. Thus by Implicit Function Theorem, there exists a neighbourhood $U$ of $(M_0, \mathbf{r}_0, \mathbf{c}_0)$ in $\mathcal{M}_{\mathfrak{P}} \times \mathfrak{D}$ and a map $\psi : U \to \mathbb{R}^n \times \mathbb{R}^m$ such that

1. $\psi(M_0, \mathbf{r}_0, \mathbf{c}_0) = (\alpha_0, \beta_0)$,

2. denote $\psi(M, \mathbf{r}, \mathbf{c}) = (\alpha, \beta)$, then the last component of $\beta$ vanishes, $\beta_m = 0$, for any $(M, \mathbf{r}, \mathbf{c}) \in U$,

3. $N((M_0, \mathbf{r}_0, \mathbf{c}_0; \psi(M_0, \mathbf{r}_0, \mathbf{c}_0)) = \mathbf{0}$, thus $\psi(M, \mathbf{r}, \mathbf{c}) = \Psi(M, \mathbf{r}, \mathbf{c}), \forall (M, \mathbf{r}, \mathbf{c}) \in U$, by strict convexity of $\mathcal{L}$ and uniqueness of $(\alpha, \beta)$,

4. $\psi \in C^k(U)$.

For the choice of $k$ is arbitrary and the choice of $(M, \mathbf{r}, \mathbf{c})$ as an interior point of $\mathcal{M}_{\mathfrak{P}} \times \mathfrak{D}$ is also arbitrary, we may see that $\Psi$ is smooth in the interior of $\mathcal{M}_{\mathfrak{P}} \times \mathfrak{D}$.

In fact, we can show that $(\mathcal{M}_{\mathfrak{P}} \times \mathfrak{D})^\circ = \mathcal{M}_{\mathfrak{P}} \times \mathfrak{D}$, thus $\Psi$ is smooth on $\mathcal{M}_{\mathfrak{P}} \times \mathfrak{D}$.

$\mathcal{M}_{\mathfrak{P}}$ is isomorphic to an open subset $(\mathbb{R}^+)^{|\mathfrak{P}|}$ of $\mathbb{R}^{|\mathfrak{P}|}$. The set $\mathfrak{D}$ is a subset of $(\mathbb{R}^+)^{n+m}$, defined by finitely many equations and strict inequalities given in [Rothblum and Schneider, 1989, Theorem 2], especially part (e): for every subset $I \subseteq \{1, 2, \ldots, n\}$ and $J \subseteq \{1, 2, \ldots, m\}$, where $M_{ij} = 0$ for all $(i,j) \in I^c \times J$ ($I^c$ is the complement of $I$), we have

$$\sum_{i \in I} r_i \geq \sum_{j \in J} c_j$$

with equality holds if and only if $M_{ij} = 0$ for all $(i,j) \in I \times J^c$. The above condition means that the conditions are either equations or strict inequalities since the pattern $\mathfrak{P}$ is fixed. Among all these constraints, set of equations $\mathcal{E}$ define a linear subspace $V(\mathcal{E})$ of $\mathbb{R}^{n+m}$ and the set of strict inequalities $\mathcal{N}$ draws an open subset $U(\mathcal{E}, \mathcal{N})$ on $V(\mathcal{E})$. And $\mathfrak{D} = (\mathbb{R}^+)^{n+m} \cap U(\mathcal{E}, \mathcal{N})$ is open in $U(\mathcal{E}, \mathcal{N})$, so $(\mathfrak{D})^\circ = \mathfrak{D}$.

(Smoothness of $\rho$:) Since $\lambda > 0$ and for each $(i,j) \in \mathfrak{P}$, $M_{ij} > 0$, then $\rho$ is smooth from the smoothness of $x^\lambda$ on $(0, \infty)$.

(Smoothness of $\mu$:) $\mu$ is the composition of exponential functions, multiplications and additions, all of which are smooth.

Thus $\Phi = \mu \circ (\rho, \Psi)$ is smooth on $\mathcal{M}_{\mathfrak{P}} \times \mathfrak{D}$.

### E.2   Calculation of gradient of $\Phi$

We make use of the decomposition $\Phi = \mu \circ (\rho, \Psi)$ to calculate the gradient of $\Phi$.

By implicit function theorem,

$$
\begin{aligned}
(\nabla_{\mathbf{r}} \Psi)_i &= \frac{\partial \Psi}{\partial \mathbf{r}_i} \\
&= -\left( \nabla_{(\alpha, \bar{\beta})} N \right)^{-1} (\nabla_{\mathbf{r}} N)_i \\
&= -\left( \nabla^2_{(\alpha, \bar{\beta})} \mathcal{L} \right)^{-1} (\nabla_{\mathbf{r}} N)_i \\
&= -\frac{1}{\lambda} \begin{pmatrix} \operatorname{diag}(\mathbf{r}) & \overline{M^{(\lambda)}} \\ \overline{M^{(\lambda)}}^\top & \operatorname{diag}(\bar{\mathbf{c}}) \end{pmatrix}^{-1} \begin{pmatrix} (\delta_{\mathbf{i}})_n \\ \mathbf{0}_{(m-1)} \end{pmatrix} \\
&= -\frac{1}{\lambda} \begin{pmatrix} \operatorname{diag}(\mathbf{r}) & \overline{M^{(\lambda)}} \\ \overline{M^{(\lambda)}}^\top & \operatorname{diag}(\bar{\mathbf{c}}) \end{pmatrix}^{-1}_{\text{col-}i}
\end{aligned}
$$

In the last equality, the subscript col-$i$ means the $i$-th column of the inverse matrix with $1 \leq i \leq n$.

$$
\begin{aligned}
(\nabla_M \Psi)_{ij} &= \frac{\partial \Psi}{\partial M_{ij}} \\
&= -\left( \nabla^2_{(\alpha, \bar{\beta})} \mathcal{L} \right)^{-1} (\nabla_M N)_{ij} \\
&= \frac{1}{\lambda} \begin{pmatrix} \operatorname{diag}(\mathbf{r}) & \overline{M^{(\lambda)}} \\ \overline{M^{(\lambda)}}^\top & \operatorname{diag}(\bar{\mathbf{c}}) \end{pmatrix}^{-1} \cdot \lambda e^{\lambda(\alpha_i + \beta_j)} M_{ij}^{\lambda-1} \begin{pmatrix} \delta_i \\ \delta_j \end{pmatrix} \\
&= \frac{M_{ij}^{(\lambda)}}{M_{ij}} \left[ \begin{pmatrix} \operatorname{diag}(\mathbf{r}) & \overline{M^{(\lambda)}} \\ \overline{M^{(\lambda)}}^\top & \operatorname{diag}(\bar{\mathbf{c}}) \end{pmatrix}^{-1}_{\text{col-}i} + \begin{pmatrix} \operatorname{diag}(\mathbf{r}) & \overline{M^{(\lambda)}} \\ \overline{M^{(\lambda)}}^\top & \operatorname{diag}(\bar{\mathbf{c}}) \end{pmatrix}^{-1}_{\text{col-}(n+\bar{j})} \right]
\end{aligned}
$$

$\bar{j}$ means that term does not exist if $j = m$.

In addition, to calculate $\left( \begin{array}{cc} \mathrm{diag}(\mathbf{r}) & \overline{M^{(\lambda)}} \\ \overline{M^{(\lambda)}}^{\top} & \mathrm{diag}(\bar{\mathbf{c}}) \end{array} \right)^{-1}$, we can use the formula

$$\left( \begin{array}{cc} A & B \\ C & D \end{array} \right)^{-1} = \left( \begin{array}{cc} M & -MBD^{-1} \\ -D^{-1}CM & D^{-1} + D^{-1}CMBD^{-1} \end{array} \right)$$

where $M = (A - BD^{-1}C)^{-1}$.

For $\rho$:

$$\frac{\partial \rho}{\partial M_{ij}} = \lambda M_{ij}^{\lambda-1} E(i,j) \tag{6}$$

with $E(i,j)$ a $n \times m$-matrix where $E(i,j)_{ij} = 1$ and all other entries vanish. And

$$\nabla_{(\mathbf{r},\mathbf{c})}\rho = \mathbf{0}. \tag{7}$$

For $\mu$:

$$\frac{\partial \mu}{\partial \alpha_i}(P,\alpha,\beta) = \lambda \mathrm{diag}(\delta_i \lambda \alpha) P \mathrm{diag}(\lambda \beta) = \lambda P^*_{(i,\_)}$$

where $P^*_{(i,\_)}$ is a matrix with $i$-th row the same as $i$-th row of $P^*$ and vanishes elsewhere.

Similarly,

$$\frac{\partial \mu}{\partial \beta_j}(P,\alpha,\beta) = \lambda \mathrm{diag}(\lambda \alpha) P \mathrm{diag}(\delta_j \lambda \beta) = \lambda P^*_{(\_,j)}$$

with $j \leq m-1$ but the size of $P^*_{(\_,j)}$ is still $n \times m$.

And

$$\frac{\partial \mu}{\partial P_{ij}} = \mathrm{diag}(\lambda \alpha) E(i,j) \mathrm{diag}(\lambda \beta) = \frac{P^*_{ij}}{P_{ij}} E(i,j)$$

where $P^*$ is the $(\mathbf{r}, \mathbf{c})$-Sinkhorn scaling limit matrix of $P$.

Finally, we can combine all the results above to calculate the gradient of $\Phi$. We will use $(\alpha, \beta)$ for $\Psi$, use $P$ for $\rho$ when it is convenient.

$$
\begin{aligned}
(\nabla_{\mathbf{r}}\Phi)_t &= \frac{\partial \Phi}{\partial \mathbf{r}_t} \\
&= \sum_{i,j=1}^{n,m} \frac{\partial \mu}{\partial \rho_{ij}} \frac{\partial \rho_{ij}}{\partial \mathbf{r}_t} + \sum_{i=1}^{n} \frac{\partial \mu}{\partial \alpha_i} \frac{\partial \alpha_i}{\partial \mathbf{r}_t} + \sum_{j=1}^{m-1} \frac{\partial \mu}{\partial \beta_j} \frac{\partial \beta_j}{\partial \mathbf{r}_t} \\
&= 0 + \sum_{i=1}^{n} \left( \frac{\partial \Psi_i}{\partial \mathbf{r}_t} \right) \frac{\partial \mu}{\partial \alpha_i} + \sum_{j=1}^{m-1} \left( \frac{\partial \Psi_{n+j}}{\partial \mathbf{r}_t} \right) \frac{\partial \mu}{\partial \beta_j}
\end{aligned}
$$

If we write the column $t$ of matrix $\left( \begin{array}{cc} \mathrm{diag}(\mathbf{r}) & \overline{M^{(\lambda)}} \\ \overline{M^{(\lambda)}}^{\top} & \mathrm{diag}(\bar{\mathbf{c}}) \end{array} \right)^{-1}$ in terms of $\left( \begin{array}{c} \mathbf{u} \\ \mathbf{v} \end{array} \right)$ with $\mathbf{u} \in \mathbb{R}^n$ and $\mathbf{v} \in \mathbb{R}^m$ with the last entry $\mathbf{v}_m = 0$ then

$$(\nabla_{\mathbf{r}}\Phi)_t = -\mathrm{diag}(\mathbf{u}) M^{(\lambda)} - M^{(\lambda)}\mathrm{diag}(\mathbf{v})$$

To calculate $\nabla_{\mathbf{c}}\Phi$, we choose an elegant way by using the above calculations. We rewrite the map $\Phi$ as $\Phi(M,\mathbf{r},\mathbf{c}) = (\Phi(M^{\vee},\mathbf{r}^{\vee},\mathbf{c}^{\vee}))^{\top}$ with $M^{\vee} = M^{\top}$, $\mathbf{r}^{\vee} = \mathbf{c}$ and $\mathbf{c}^{\vee} = \mathbf{r}$. The transpose of $M$, after regularization, scaled to $(\mathbf{c},\mathbf{r})$ is exactly $(M^{(\lambda)})^{\top}$.

So we have $\nabla_{\mathbf{c}}\Phi(M,\mathbf{r},\mathbf{c}) = \nabla_{\mathbf{r}^{\vee}}(\Phi(M^{\vee},\mathbf{r}^{\vee},\mathbf{c}^{\vee}))^{\top}$, thus

$$
\begin{aligned}
(\nabla_{\mathbf{c}}\Phi(M,\mathbf{r},\mathbf{c}))_s &= ((\nabla_{\mathbf{r}^{\vee}}\Phi(M^{\vee},\mathbf{r}^{\vee},\mathbf{c}^{\vee}))_s)^{\top} \\
&= -M^{(\lambda)}\mathrm{diag}(\mathbf{u}) - \mathrm{diag}(\mathbf{v})M^{(\lambda)},
\end{aligned}
$$

where $\begin{pmatrix} \mathbf{u} \\ \bar{\mathbf{v}} \end{pmatrix}$ is the $s$-th column of matrix $\begin{pmatrix} \text{diag}(\mathbf{c}) & \overline{M^{(\lambda)}}^{\top} \\ \overline{M^{(\lambda)}} & \text{diag}(\bar{\mathbf{r}}) \end{pmatrix}^{-1}$.

At last,

$$
\begin{aligned}
(\nabla_M \Phi)_{st} &= \frac{\partial \Phi}{\partial M_{st}} \\
&= \sum_{i,j=1}^{n,m} \frac{\partial \mu}{\partial \rho_{ij}} \frac{\partial \rho_{ij}}{\partial M_{st}} + \sum_{i=1}^{n} \frac{\partial \mu}{\partial \alpha_i} \frac{\partial \alpha_i}{\partial M_{st}} + \sum_{j=1}^{m-1} \frac{\partial \mu}{\partial \beta_j} \frac{\partial \beta_j}{\partial M_{st}} \\
&= \lambda \frac{M_{st}^{(\lambda)}}{M_{st}} \left( E(s,t) - \text{diag}\left(\mathbf{u}\right) M^{(\lambda)} - M^{(\lambda)} \text{diag}\left(\mathbf{v}\right) \right)
\end{aligned}
$$

where $\mathbf{u} \in \mathbb{R}^n$, $\mathbf{v} \in \mathbb{R}^m$ with the last entry $\mathbf{v}_m = 0$, and

$$
\begin{pmatrix} \mathbf{u} \\ \bar{\mathbf{v}} \end{pmatrix} = \left[ \begin{pmatrix} \text{diag}(\mathbf{r}) & \overline{M^{(\lambda)}} \\ \overline{M^{(\lambda)}}^{\top} & \text{diag}(\bar{\mathbf{c}}) \end{pmatrix}^{-1}_{\text{col-}s} + \begin{pmatrix} \text{diag}(\mathbf{r}) & \overline{M^{(\lambda)}} \\ \overline{M^{(\lambda)}}^{\top} & \text{diag}(\bar{\mathbf{c}}) \end{pmatrix}^{-1}_{\text{col-}(n+\bar{t})} \right],
$$

for $(s,t) \in \mathfrak{P}$, and $\bar{t}$ means that term does not exist if $t = m$.

$\square$