[Reviews · NeurIPS 2020]

Review 1

Summary and Contributions: The paper offers a mathematical interpretation of the problem of cooperative communication as a problem of optimal transport. The proposed framework is shown to apply effectively to existing problems. This general formalization is then exploited to derive some theoretical properties for cooperative communication.

Strengths: The proposed formalism is extremely elegant, and the paper succeeds in showing how drawing a parallel between the two domains can yield theoretical foundations for existing domains in a quite simple way. The presented results are theoretically sound and rely on strong mathematical theories. To my knowledge, using the formalism of OT for cooperative communication is novel and could interest a certain audience from the NeurIPS community.

Weaknesses: I said a certain audience, but I would say the paper is maybe a bit difficult to read for someone from NeurIPS community. The thing is that it relates extremely briefly many notions from different domains which are not of direct interest for machine learners, and that these domains are not well described. The paper would clearly benefit from giving a proper definition of the different problems it addresses, including cooperative communication which is not described before a quarter of the paper! Concerning the novelty of the contribution, the authors claim that there is no strong mathematical framework for cooperative communication. However, I think this claim is a bit strong since it seems to ignore completely game-theoretic analysis of communication games. I suppose that the authors simply meant that their theory is the first one to unify all these works, but I am a bit concerned by this formulation. Finally, I would have expected from the authors a discussion on whether the proposed theory can have practical consequences or is just a matter of unifying multiple problems under a unique mathematical formulation (which is already a goal in se). Considering the response of the authors, I would add several comments. About the relevance to the NeurIPS community: My comment has been misunderstood, probably because of an ambiguous formulation of mine, for which I sincerely apologize. I am convinced that the paper is relevant for the NeurIPS community (which I state in the strengths). What I really meant here is that the paper does not, in its current form, take the time to show direct connections to the domains it addresses. The effort of seeing a connection is up to the reader. If the applicative domains are relevant for some communities in machine learning, the paper is advertised around an idea which I think is not that familiar to most machine learners. It is noticeable that the term “cooperative communication” is not properly defined in the first pages, and that the paper considers the reader as familiar with this terminology. From the first two pages for instance, I would not be sure that a specialist of machine teaching would see a direct connection with their field, which is however the case. In this direction, I think that the summary provided in the response is the kind of text which would make these connections explicit and greatly help the reading of the paper. About the practical applications: I do not have any doubts that theory has direct application. However, I think it is important, especially in a paper of this kind, that this point is more thoroughly discussed. The remarks given in the responses are a very good start. As a conclusion, I think the paper is excellent and proposes an impactful vision. My main concern is that the paper, in its current form, makes this impact quite hidden to those it should concern. In case of acceptation, I strongly encourage the authors to work on this, so that this research gets the impact it deserves, in particular onto communities which might ignore the paper if published in the current form.

Correctness: The results presented in the paper are sound.

Clarity: The main problem of the paper is its overall lack of clarity. It is difficult to understand what the main claim is and how all parts relate to each other. The paper lacks a clear summary of the main justifications and advantages of using this formulation of cooperative communication as OT. It also lacks explicative examples throughout the paper to have a better understanding of the proposed applications (even if some are present in the supplementary material). The proposed summary is an excellent first step in the direction of clarifying the paper.

Relation to Prior Work: The paper is based on a unification of various researches under a unifying mathematical framework. It seems very complete in terms of the covered references. However, as mentioned above, I think it lacks a proper discussion on the game-theoretic literature on cooperation. If such a literature does not have the same impact as the paper, this should be stated in any case.

Reproducibility: Yes

Additional Feedback: see previous comments Very minor comment: Due to the nature of the main domain of the paper, I think the statement of broader impact could be a bit more detailed. I changed my score to 6, mainly because I think the summary proposed in the response is a very promising direction for making the paper clearer (see reply in the weakness part)


Review 2

Summary and Contributions: The authors propose a mathematical theory of cooperative communication using optimal transport. The derivations show that the prior Bayesian approaches are the special case of this theory when \lambda=1, and other related works are also the special case as they only compute one or two steps of the Sinkhorn scaling.

Strengths: + It's a very nice application of optimal transport, which perfectly unifies the prior methods of cooperative communication. The deviations are very thorough. + Detailed examples make it easy to follow

Weaknesses: My biggest concern is the lack of experiments to really showcase the usefulness of this new unified model. There are several challenging tasks previously used in the ML community; they often find it difficult to solve using I-POMDP, since recursions of multi-agent would make the problem intractable due to large space. Additionally, I'd like to see whether this new framework would enable the agents to emerge some behaviors during cooperative communication, as shown in [1]. Since it's a unified model, it should be able to account for all the previously demonstrated emergent behaviors. [1] Emergence of Grounded CompositionalLanguage in Multi-Agent Populations, AAAI 2018

Correctness: See weakness

Clarity: The paper is overall well-written and easy to follow

Relation to Prior Work: See weakness

Reproducibility: Yes

Additional Feedback:


Review 3

Summary and Contributions: This paper presents a unifying framework for analyzing cooperative communication, such as teaching or pragmatic linguistic inference. By establishing a connection to the theory of optimal transport, the paper makes a link between an important topic of research in cognitive science and robotics and an extensive mathematical literature. Drawing on this connection, it's possible to establish rigorous conditions under which cooperative communication converges and the limitations of some current approximations.

Strengths: I think this is potentially the definitive paper on the mathematical analysis of cooperative communication, establishing a firm foundation for this topic and providing a lot of ground on which to build. Several analyses demonstrate the utility of this framework for answering basic questions about cooperative communication, including robustness and the quality of approximations. The paper presents both theoretical results and a comparison to human data, which shows that the optimal solution provides a better fit than standard approximations. These results are likely to be of interest to researchers in both cognitive science and robotics who are using these theories.

Weaknesses: 1. Connections to optimal transport and Sinkhorn-Knopp have already been made in the literature, in the following papers (which are cited). My reading is that this is still a genuine advance, making the connection to OT explicit and drawing out its consequences. However, the relationship with this previous work should be spelled out. https://arxiv.org/abs/1810.02423 https://arxiv.org/pdf/1705.08971.pdf 2. There are a lot of moving parts in the formal framework -- entropy regularization etc. -- and the basic establishment of equivalence doesn't seem to draw on all of those. Given that there are variations in the different cognitive science frameworks, such as use of costs in RSA but not optimal pedagogy, it would be worth spelling out the actual formal correspondences to each of these cases and identifying which components of EOT need to have what values to produce that. Maybe a Table or more details in the supplement? 3. The information theoretic part in 2.4 seems to come out of nowhere and it's not clear what work it's doing. This needs to be better contextualized. 4. I really liked the comparison to human data, I thought this was one of the strongest results. Given the motivation, it would be nice to see the other results tied more closely to human behavior. For an audience of cognitive scientists, showing that there are a clear implications of this equivalence for modeling behavior is going to be a big factor in impact. The connection to information to

Correctness: Yes

Clarity: Yes, modulo points above

Relation to Prior Work: No, the relationship to the two papers indicated above needs to be made more explicit.

Reproducibility: Yes

Additional Feedback:

[Author Response · NeurIPS 2020]

We would like to thank the reviewers for their generous comments and suggestions. We are encouraged that they found that our formalism is *extremely elegant* (R2) and *perfectly unifies the prior methods* (R3), and that our analyses were *sound and thorough* (R2, R3). Reviewer 4's assessment that **"this is potentially the definitive paper on the mathematical analysis of cooperative communication, establishing a firm foundation for this topic and providing a lot of ground on which to build."** is worthy of special note. We believe the strength of these endorsements support acceptance of the paper in NeurIPS, and encourage the reviewers to consider our responses in their decisions.

**Summary.** Recall that we present a unifying EOT framework for analyzing cooperative communication. Build upon machinery in optimal transport, we provide answers to fundamental questions about a broad **class of models** of cooperative communication. Specifically, we (a) theoretically guarantee the existence of optimal communication plan and algorithmically ensure the achievability of such plans by convergence of Sinkhorn scaling; (b) mathematically analyze and computationally verify the robustness to violations of common ground. These theoretical results are important because they establish viability in practice. It is implausible (even impossible) for any two agents to have exactly the common ground, if any model is to serve as a theory of human or machine behavior in realistic settings, besides the existence of optimal plans, the stability result is necessary. Moreover, our framework also provides strong links over a wide range of research topics—pragmatic reasoning, robotics, machine teaching, and Bayesian probability—that are important to the NeurIPS community.

(R4-Q1) "What is the relation between the current work and the literatures on cooperative inference [Yang et al., 2018, Wang et al., 2019]?" **Cooperative inference is a special case of our unified EOT framework** with greedy parameter $\lambda = 1$ (Prop. 3 in Sec.2.3, see detail derivation in Supplement Sec. B.1). The new, general formulation has several theoretical and practical implications. Theoretically, previously unproven results include: convergence of cooperative communication for *arbitrary* priors (Prop. 3), smoothness (in particular differentiability) of cooperative communication which implies the deviations in common ground are repairable (Prop.4& 5), analysis of instability under greedy data selection with difference choice of $\lambda$ (Sec.3.3, Prop 6).

(R2) "[W]hether the proposed theory can have practical consequences or just a matter of unifying..." **Fundamentally, theoretical guarantees—of convergence, of algorithms, of robustness—have deeply practical implications**. For example, existing models assume, without basis, that models will work despite the fact that communicating partners can never have perfect common ground. Our analysis justifies practical application of the class of models, and suggests under when (in terms of parameterizations) and why violations may result in failures of communication.

(R4-Q2) "Maybe a table that spells out the formal correspondence between various existing cognitive models and components of EOT"? **Great suggestion!** We will include a such table in the revised supplemental materials.

(R4-Q3) "Connection with the information theoretic part in 2.4 needs to be better contextualized". **We will do so.** Cooperative communication, like standard information theory, involves communication over a channel. It is therefore interesting and important to ask whether there is a formal connection. Notably, the connection established in Sec.2.4 is an additional demonstration of the value of our general formulation.

(R3,R4-Q4) " [L]ack of experiments to really showcase the usefulness of this new unified model" &"[I]t would be nice to see the other results tied more closely to human behavior." **Our goal is establishing a mathematical framework for proving statements about the class of models**, rather than assessing a single model. To that end, our theoretical results are novel in both the machine learning and human learning literatures. Our simulation and model fitting results illustrate the implications of our analyses (Sec.3) by demonstrating predicted differences among specific models for both machine and human cooperative communication (Sec.4).

(R2, R3) "[I]gnore completely game-theoretic analysis of communication games" & "enable the agents to emerge some behaviors during cooperative communication, as in [1]." **We will acknowledge game theory**; however, we are unaware of game theoretic approaches that are competitive in both human and machine cooperative communication. **Our mathematical analysis is of single interactions, unlike POMDPs as in [1]**. Note that [1] has no proofs.

(R2) "[D]omains not of direct interest to machine learners". **We respectfully disagree, and we provide evidence.** The idea of cooperative communication has been proposed and applied in a wide range of existing machine learning models such as teaching by demonstration [Ho et al., 2016] and cooperative inverse reinforcement learning [Hadfield-Menell et al., 2016] which were both published in NeurIPS. In addition, NeurIPS has published papers on machine teaching (5 papers in NeurIPS 2019!), and machine-human or machine-machine collaboration and teaming. Our framework also bridges many topics central to the NeurIPS community (as measured by appearances in titles of papers): optimal transport (13 papers last year), Sinkhorn algorithm (3 papers) and Bayesian (40 papers) probability (14 papers).

We greatly appreciate the reviewers' enthusiastic endorsements, as well as their suggestions and comments. We will reorganize the contents and provide clear summary of the main justifications and advantages of the EOT framework to incorporate reviewers feedback.

[Meta-Review · NeurIPS 2020]

Three knowledgeable referees all concur that the paper should be accepted. This paper presents a unifying framework for analyzing cooperative communication (drawing from ideas in optimal transport). The paper establishes "a firm foundation for this topic and" provides ideas to build from. I concur with accepting the paper.